# Vulnerability of progeroid smooth muscle cells to biomechanical forces is mediated by MMP13

Patricia R. Pitrez [1,2], Luís Estronca[1,2], Luís Miguel Monteiro [1], Guillem Colell[3], Helena Vazão[1], Deolinda Santinha [1,2], Karim Harhouri [4], Daniel Thornton[5], Claire Navarro [4,6], Anne-Laure Egesipe[7], Tânia Carvalho [8], Rodrigo L. Dos Santos[9], Nicolas Lévy[4,10], James C. Smith [11], João Pedro de Magalhães[1,5], Alessandro Ori [12], Andreia Bernardo [11], Annachiara De Sandre-Giovannoli [4,10,13], Xavier Nissan [7], Anna Rosell[3] & Lino Ferreira [1,2✉]

Hutchinson-Gilford Progeria Syndrome (HGPS) is a premature aging disease in children that leads to early death. Smooth muscle cells (SMCs) are the most affected cells in HGPS individuals, although the reason for such vulnerability remains poorly understood. In this work, we develop a microfluidic chip formed by HGPS-SMCs generated from induced pluripotent stem cells (iPSCs), to study their vulnerability to flow shear stress. HGPS-iPSC SMCs cultured under arterial flow conditions detach from the chip after a few days of culture; this process is mediated by the upregulation of metalloprotease 13 (MMP13). Importantly, double-mutant $Lmna^{G609G/G609G}Mmp13^{-/-}$ mice or $Lmna^{G609G/G609G}Mmp13^{+/+}$ mice treated with a MMP inhibitor show lower SMC loss in the aortic arch than controls. MMP13 upregulation appears to be mediated, at least in part, by the upregulation of glycocalyx. Our HGPS-SMCs chip represents a platform for developing treatments for HGPS individuals that may complement previous pre-clinical and clinical treatments.

[1] Center for Neuroscience and Cell Biology, University of Coimbra, Coimbra, Portugal. [2] Faculty of Medicine, University of Coimbra, Coimbra, Portugal. [3] Neurovascular Research Laboratory, Vall d'Hebron Research Institute, Universitat Autònoma de Barcelona, Passeig Vall d'Hebron 119-129, 08035 Barcelona, Spain. [4] Aix Marseille Univ, INSERM, MMG, Marseille, France. [5] Integrative Genomics of Ageing Group, Institute of Ageing and Chronic Disease, University of Liverpool, Liverpool L7 8TX, UK. [6] Progelife, Marseille, France. [7] CECS, I-STEM, AFM, Institute for Stem Cell Therapy and Exploration of Monogenic Diseases, Evry Cedex, France. [8] IMM, Instituto de Medicina Molecular, Universidade de Lisboa, Lisbon, Portugal. [9] 25 Cambridge Science Park, Mogrify Ltd, Milton Road, Cambridge CB4 0FW, UK. [10] Molecular Genetics Laboratory, Department of Medical Genetics, La Timone Children's Hospital, Marseille, France. [11] Developmental Biology Laboratory, Francis Crick Institute, London NW1 1AT, UK. [12] Leibniz Institute on Aging - Fritz Lipmann Institute, 07745 Jena, Germany. [13] CRB Assistance Publique des Hôpitaux de Marseille (CRB AP-HM, TAC), Marseille, France. ✉email: lino@uc-biotech.pt

Hutchinson–Gilford Progeria Syndrome (HGPS) is caused by a single mutation in the lamin A/C gene (LMNA), resulting in the generation of an abnormal lamin A precursor named progerin[1,2]. One of the key reasons of premature death is the loss of smooth muscle cells (SMCs) in the medial layer of large arteries, followed by the appearance of collagen and extracellular matrix (ECM) and the development of a severe arteriosclerotic process that leads to increased arterial stiffness[3–5]. The reasons of SMC loss remain to be determined. It has been suggested that this may happen due to pathophysiological changes inherent to prelamin A/progerin accumulation, such as the acceleration of vascular calcification via the activation of the DNA damage response and senescence-associated secretory phenotypes in vascular SMCs[6] or the downregulation of PARP1[7]. It has also been shown that the combined effect of progerin accumulation and mechanical stress in mouse SMCs over-expressing progerin promoted cell detachment and death, while the disruption of the linker between nucleoskeleton and cytoskeleton complex ameliorated the toxic effects of progerin[8]. Neither of these studies have fully addressed the reasons behind SMC detachment and thus which therapeutic approach could be effective to prevent SMC loss.

Induced pluripotent stem cells (iPSCs) offer an unlimited source of SMCs to study HGPS. Recent studies have generated iPSCs from fibroblasts obtained from individuals with HGPS (hereafter referred to as HGPS-iPSCs)[9–11]. Strikingly, HGPS-iPSCs show low lamin A/C and progerin protein expression in the pluripotent state. However, the expression of progerin is reactivated after HGPS-iPSC differentiation into SMCs[7,9]. The differentiated cells show nuclear dysmorphology, cell growth retardation, susceptibility to apoptosis, proliferation reduction, and DNA-repair defects; however, SMC performance under flow conditions has not been evaluated.

In this work, we develop an in vitro cell system comprising SMCs derived from HGPS-iPSCs cultured under flow conditions in a microfluidic device. We identify MMP13 as a mediator of SMC detachment using chemical and genetic assays. The generated double-mutant $Lmna^{G609G/G609G}Mmp13^{-/-}$ mice show an increase in SMCs in the aortic arch and a decrease in progerin-positive cells. In addition, the inhibition of MMP13 in $Lmna^{G609G/G609G}$ mice by Batimastat, a drug that has been previously tested in clinical trials in cancer patients, reduces SMC loss. The results present here open perspectives for HGPS treatment.

## Results

### SMCs derived from HGPS-iPSCs are functional and share similar features to progerin-expressing cells.

iPSCs were generated from HGPS skin fibroblasts and characterized as previously described[10]. iPSCs generated from non-disease cells (N-iPSCs), HGPS skin fibroblasts, and non-disease somatic human vascular smooth muscle cells (hVSMCs) were used as controls. The mutation in the LMNA gene, both in HGPS skin fibroblasts and HGPS-iPSCs, was confirmed by Sanger sequencing (Supplementary Fig. 1). As expected, undifferentiated HGPS-iPSCs expressed low levels of HGPS markers, such as progerin, as well as low levels of SMC markers, such as α-SMA and SMα-22[12,13] (Supplementary Fig. 2a). To induce the differentiation of HGPS-iPSCs or N-iPSCs into SMCs, CD34+ cells were isolated by magnetic-activated cell sorting from embryoid bodies (EBs) cultured for 10 days in suspension (Fig. 1a)[14]. At this stage, HGPS-CD34+ cells already express higher levels of progerin mRNA transcripts relative to N-iPSCs but relatively low levels of SMC mRNA transcripts compared with somatic hVSMCs (Supplementary Fig. 2b). HGPS-CD34+ cells were then cultured in SMC induction media (Supplementary Fig. 3) followed by SMC maturation media (Supplementary Fig. 4) for an additional four passages. Matured SMCs are referred to as HGPS-iPSC SMCs or N-iPSC SMCs based on their phenotype, genotype, and functional properties (see below). Both HGPS-iPSC SMCs and N-iPSC SMCs have similar or higher expression of SMC mRNA transcripts than somatic hVSMCs (Supplementary Fig. 4a). Greater than 95% of both differentiated cells express α-SMA, smooth muscle myosin heavy chain (SMMHC), and calponin proteins (Fig. 1b). Moreover, HGPS-iPSC SMCs express progerin mRNA transcripts (Fig. 1c) and progerin protein (Supplementary Fig. 4b, c). Similar results were obtained for SMCs derived from HGPS-iPSCs generated from a second Progeria individual; however, the differentiated cells showed higher progerin protein levels than the first Progeria individual (Supplementary Fig. 5). Importantly, HGPS-iPSC SMCs and N-iPSC SMCs are functional as they respond to vasoactive agents such as histamine and angiotensin (Supplementary Fig. 4d) and they contract after exposure to carbachol (Supplementary Fig. 4e).

SMCs derived from HGPS-iPSCs share similar features to progerin-expressing cells. Cell lines forced to express progerin show the activation of several NOTCH signaling pathway effectors[15]. Indeed, our results showed that HGPS-iPSC CD34+ cells had higher expression of NOTCH signaling pathway mRNA transcripts than N-iPSC CD34+ cells (Supplementary Fig. 6). Mature HGPS-iPSC SMCs also expressed higher levels of NOTCH ligand and receptors than N-iPSC SMCs (Supplementary Fig. 6a). In addition, HGPS-iPSC SMCs responded to farnesyltransferase inhibitors, as has been shown in other Progeria cell models[16–18]. In the current work, HGPS-iPSC SMCs treated with lonafarnib for 48 h accumulated nuclear prelamin A and showed a decrease in nuclear shape abnormalities and nuclear blebbing (Supplementary Fig. 7a–c). Taken together, the cells differentiated from HGPS-iPSCs-expressed SMC and progeroid markers, are functional and exhibit physiological responses.

### HGPS-iPSC SMCs are vulnerable to arterial shear stress.

SMCs differentiated from N-iPSCs or HGPS-iPSCs were seeded in a microfluidics system and cultured under flow conditions for up to 7 days (Fig. 1d). Because SMCs from large arteries are the most affected in blood vessels in HGPS, we used a flow of 20 dyne/cm$^2$, which is typically found in arterial blood vessels[19]. N-iPSC SMCs (Fig. 1g), hVSMCs, or HGPS fibroblasts (80% of which express progerin) (Fig. 1e, g) can be cultured in the microfluidics system for at least 7 days without a visible loss in cell number. In contrast, HGPS-iPSC SMCs cultured under flow conditions formed cell clumps overtime (Fig. 1f), and most of the cells detached from the substrate at day 4 as confirmed by cell number (Fig. 1g) and metabolic analyses (Fig. 1h). During this time period, the percentage of cells expressing progerin and displaying nuclear abnormalities increased significantly until day 4 (Supplementary Fig. 8). Our results indicate that SMC detachment is mediated by progerin accumulation, as the inhibition of progerin by antisense morpholinos[20] significantly decreased HGPS-iPSC SMC detachment (Supplementary Fig. 9). In addition, we showed that HGPS-iPSC SMCs with high progerin expression (30% of the cells express progerin at day 0) detached from the surface of the microfluidics system in a short time (<12 h) (Supplementary Fig. 5g). To confirm that progerin accumulation is responsible for SMC loss, a frameshift mutant stem cell line was generated (HGPSΔ2-iPSCs) to knockout the HGPS mutant allele and generated a disease cell line, as previously described in the mouse[21] (Fig. 2a and Supplementary Fig. 10). Specifically, a two-base pair deletion on exon 11, upstream of the HGPS point mutation (1814C>T), was generated. Notably, HGPSΔ2-iPSCs expressed

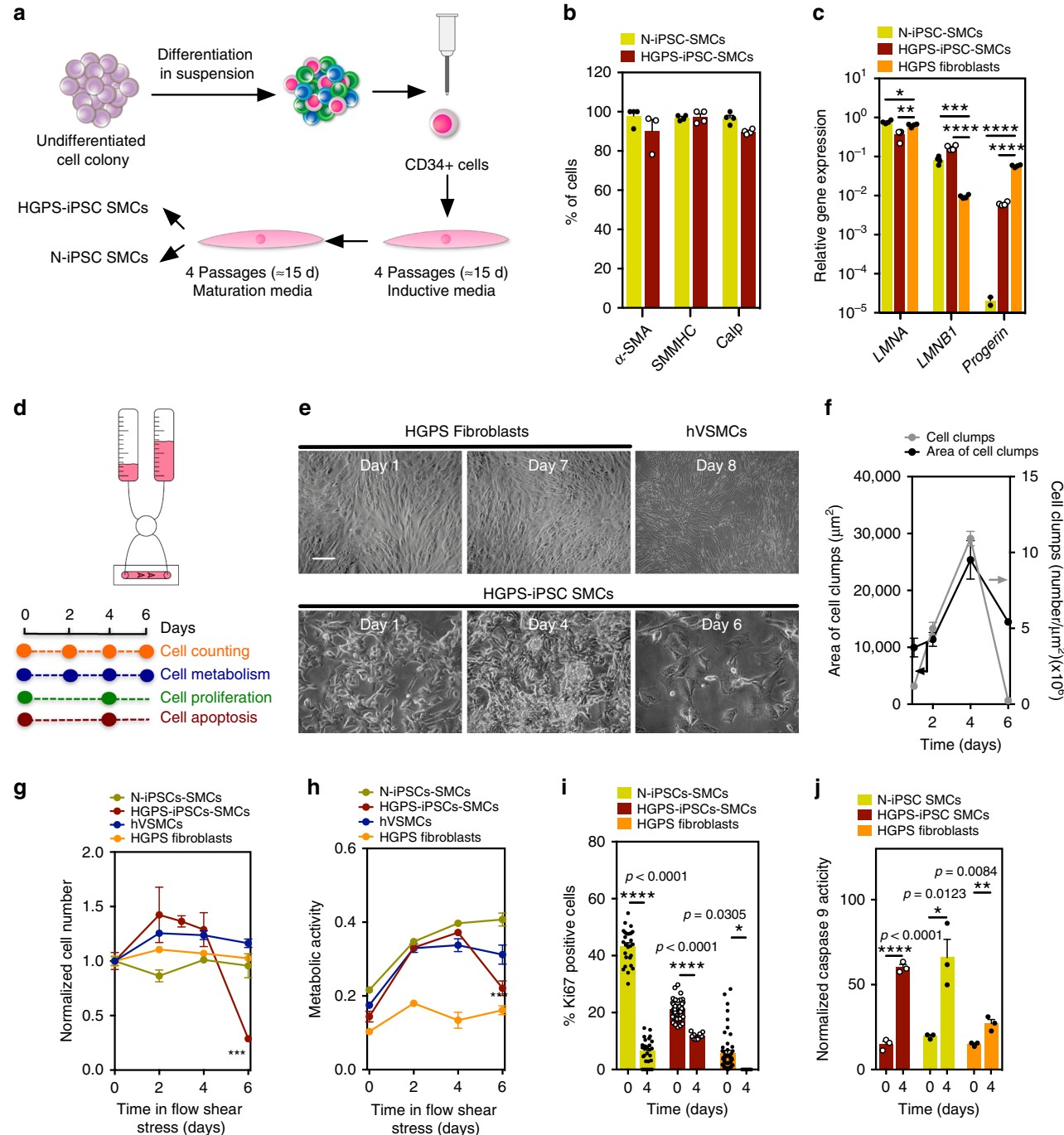

little or no progerin upon differentiation into SMCs as demonstrated at the transcript and protein levels and did not detach under flow culture conditions (Fig. 2).

HGPS-iPSC SMC detachment does not seem to be mediated by cell apoptosis. Before cell detachment, HGPS-iPSC SMCs showed: (i) poor proliferation (as monitored by Ki67 staining) confirming their contractile phenotype (Fig. 1i), (ii) similar levels of apoptosis as N-iPSC SMCs as confirmed by caspase-9 activity (Fig. 1j), (iii) an osteogenic differentiation program (Supplementary Fig. 11a, b), (iv) increased DNA damage[6] (Supplementary Fig. 12), and (v) downregulation of NOTCH[15,22] (Supplementary Fig. 13) signaling pathways. Because the in vivo shear stress from blood flow is not directly sensed by SMCs but by endothelial cells (ECs), we co-cultured SMCs differentiated from HGPS-iPSCs (directly attached to the microfluidics substrate) with human umbilical

artery ECs (HUAECs, on top of the SMCs) under flow conditions. Initially, we screened different culture conditions and we found that endothelial growth media-2 (EGM-2) medium was a suitable medium to support both cells (Supplementary Fig. 14). Then, we co-cultured HUAECs and HGPS-iPSC SMCs at different ratios (1.6, 1, and 0.6) under flow conditions. In all the ratios tested, we had a monolayer of HUAECs (Supplementary Fig. 15a) and HGPS-iPSC SMCs at time zero. After 6 days in flow conditions, a significant percentage (>40%) of HGPS-iPSC SMCs was lost (Supplementary Fig. 15b). For the highest ratio tested (1.6), the loss of HGPS-iPSC SMCs occurred without visible loss of ECs. Yet, for EC:SMC ratios below 1, part of ECs also detached from the microfluidic chamber indicating that, a low ECs density, may turn ECs vulnerable to flow conditions. Importantly, cell vulnerability to flow conditions was only observed in co-cultures

**Fig. 1 Vulnerability of HGPS-iPSC SMCs to arterial flow conditions. a** Schematic representation of the methodology used to differentiate iPSCs into SMCs. **b** Expression of SMC markers on iPSC-derived SMCs. Percentage of positive cells expressing SMC markers as evaluated by immunofluorescence (at least 100 cells were counted per each marker). Results are mean ± SEM ($n = 3$ independent experiments). **c** Expression of progeria markers on iPSC-derived SMCs. Gene expression by qRT-PCR (gene expression was normalized by the housekeeping gene *GAPDH*). HGPS fibroblasts were used as control. Results are mean ± SEM ($n = 4$ technical replicates from a pool of three independent experiments). *, **, ***, **** denote statistical significance ($p < 0.05$, $p < 0.01$, $p < 0.001$, $p < 0.0001$). Statistical analyses were performed by one-way ANOVA followed by Newman–Keuls's post test. **d** Schematic representation of the protocol used. Cells were cultured for 6–8 days in arterial flow conditions (20 dyne/cm²). **e** Light microscopy images of HGPS fibroblasts, hVSMCs, or HGPS-iPSC SMCs (10% of the cells accumulate progerin protein) at different culture days. Only HGPS-iPSC SMCs detached from the microfluidic system at day 4. Scale bar is 50 μm. **f** Number and area of cell clumps in HGPS-iPSC SMCs at different times (at least two images (×10) have been quantified per time). For area of cell clumps $n > 2$ images examined over three independent experiments; for cell clumps, $n = 3$ independent experiments. **g** Number of cells per surface area (mm²) during cell culture under arterial flow (at least three images (×10) have been quantified per time; $n = 3–7$ independent experiments). Cell number was normalized by the number of cells present at day 0. **h** Cell metabolism evaluated by the Presto Blue assay. Absorbance at 570 nm was measured and normalized to the 600-nm values for the experimental wells. $n = 3$ independent experiments. **i** Expression of nuclear proliferation marker, Ki67 (at least three images (×10) have been quantified per time). The percentage of Ki67 positive cells was evaluated by immunofluorescence. $n > 3$ images examined over three independent experiments. **j** Cell apoptosis evaluated by caspase-9 activity. Results were normalized by cell number. $n = 3$ independent experiments. From **c** to **g**, results are mean ± SEM. *, **, ***, **** denote statistical significance ($p < 0.05$, $p < 0.01$, $p < 0.001$, $p < 0.0001$). Statistical analyses were performed by a two-tailed unpaired Student's $t$ test **i** and **j**.

of HGPS-iPSC SMCs but not N-iPSC SMCs (Supplementary Fig. 15c).

It has been shown that a knock-in mouse line carrying a homozygous Lmna c.1827C>T;p.Gly609Gly mutated allele ($Lmna^{G609G/G609G}$) recapitulates most of the described alterations associated with HGPS, including the loss of SMCs[20]. Thus, to validate the results obtained for the HGPS-iPSC SMCs, we isolated SMCs from wild-type (WT mSMC) and homozygous $Lmna^{G609G/G609G}$ (HOZ mSMC) mice. Both cells expressed calponin and α-SMA, while HOZ mSMCs, but not WT SMCs, showed dysmorphic nuclei and nuclear blebbing (Fig. 3a, b). WT mSMCs were cultured under flow conditions (120 dyne/cm² to mimic mice arterial flow shear stress[23,24]) for up to 26 days without visible loss of cells (Fig. 3c). In contrast, HOZ mSMCs detached from the substrate after 8–9 days. These results confirm that HOZ mSMCs are vulnerable to flow shear stress similar to HGPS-iPSC SMCs. Overall, our results indicate that HGPS-iPSC SMCs are vulnerable to flow shear stress, as in the case of SMCs isolated from mice carrying a HGPS-like mutation in the *Lmna* gene.

**HGPS-iPSC SMCs have significant changes in extracellular matrix (ECM) secretion and MMP expression.** To gain insights into the mechanism behind SMC detachment, we performed microarray analyses on HGPS-iPSC SMCs and N-iPSC SMCs at days 0 and 4 (before cell detachment). At day 0, 2084 genes were differentially expressed (Log2FC ≥ 1; $p < 0.05$) in HPGS-iPSC SMCs vs. N-iPSC SMCs. Of these genes, 51 genes were associated with cell senescence, as determined by the intersection of all the differentially expressed genes with the CellAge database[25] (279 genes) (Supplementary Data 1). At the protein levels, HGPS-iPSC SMCs expressed higher levels of p21 and SA-β-galactosidase than N-iPSCs-SMCs and the level of senescence markers increased after culture of HGPS-iPSC SMCs in flow conditions (Supplementary Fig. 16a and Supplementary Data 5). We next performed pathway analysis on the differentially expressed genes from HGPS-iPSC SMCs at day 0 vs. day 4 (Supplementary Fig. 17 and Supplementary Data 2, 3). In general, ECM activation, secretion, and cell adhesion pathways were upregulated, whereas cell cycle and DNA replication pathways were downregulated under arterial flow conditions at day 4. Among the fifty-seven genes that were at least threefold down- or upregulated compared with day 0 ($p < 0.001$) (Fig. 4a), five were related to ECM secretion (*COL6A3, IBSP, BGN, SGCG, and EPPK1*) and one to metalloproteases (*MMP13*). The expression of these genes, as well as others, was

confirmed by qRT-PCR (Fig. 4a), and the molecular network of genes that were differentially expressed between days 0 and 4 in the HGPS-SMCs was examined by Ingenuity Pathway Analysis (Supplementary Fig. 17). Interestingly, pathway analysis suggested that *MMP13* is either a direct or indirect target of multiple genes upregulated at day 4. Moreover, *MMP13* transcript levels are elevated in HGPS-iPSC SMCs when compared with SMCs generated from the attenuated disease version of this line (HGPSΔ2-iPSC SMCs), specially post shear stress (Fig. 2g).

To further explore the gene array results, we evaluated whether the presence of ECM secreted by hVSMCs could prevent the detachment of HGPS-iPSC SMCs under arterial flow conditions. Thus, we cultured HGPS-iPSC SMCs on decellularized ECM deposited by hVSMCs or directly on top of mitotically inactivated hVSMCs (Supplementary Fig. 18). Both conditions were unable to prevent HGPS-iPSC SMC detachment. Next, we tested whether conditioned media collected from HGPS-iPSC SMCs in flow conditions for 4 days could induce the detachment of flow shear stress-insensitive hVSMCs (Fig. 4b). Surprisingly, hVSMCs detach after perfusion with HGPS-iPSC SMC-conditioned media but not with N-iPSC SMC-conditioned media (Fig. 4c). Following these results and given that *MMP13* appears to be the downstream effector for the genes misregulated at day 4 (Supplementary Fig. 17b) we decided to quantify the concentration of MMP13 in HGPS-iPSC SMC and N-iPSC SMC culture media after flow shear stress. Remarkably, MMP13 levels increased 30-fold in the HGPS-iPSC SMC culture media, but not in the control cell culture media (Fig. 4d). Similarly, higher MMP13 levels were observed in media collected from HOZ mSMCs under flow shear stress, when compared with media from WT mSMCs (Fig. 3d). Because MMP13 is produced by cells as an inactive form (proMMP13), which is then activated by cell membrane MMPs, namely MMP14 (also called MT1-MMP) and MMP2 (also called gelatinase A)[26], the catalytic activity of MMP13 secreted by HGPS-iPSC SMCs was analyzed (Supplementary Fig. 19). The concentration of proMMP13 and active MMP13 increased approximately eight- and five-fold, respectively, in culture media of HGPS-iPSC SMCs cultured in flow conditions from day 0 to day 4. Moreover, the concentration of proMMP13 and active MMP13 in cell culture media collected from N-iPSC SMCs cultured in flow conditions for 4 days was more than fourfold lower than the one observed with HGPS-iPSC SMCs. Altogether, our results indicate that HGPS-iPSC SMCs cultured under flow conditions showed increased cell senescence, ECM activation, secretion, and cell adhesion pathways upregulation and dysregulation in the expression of MMP13.

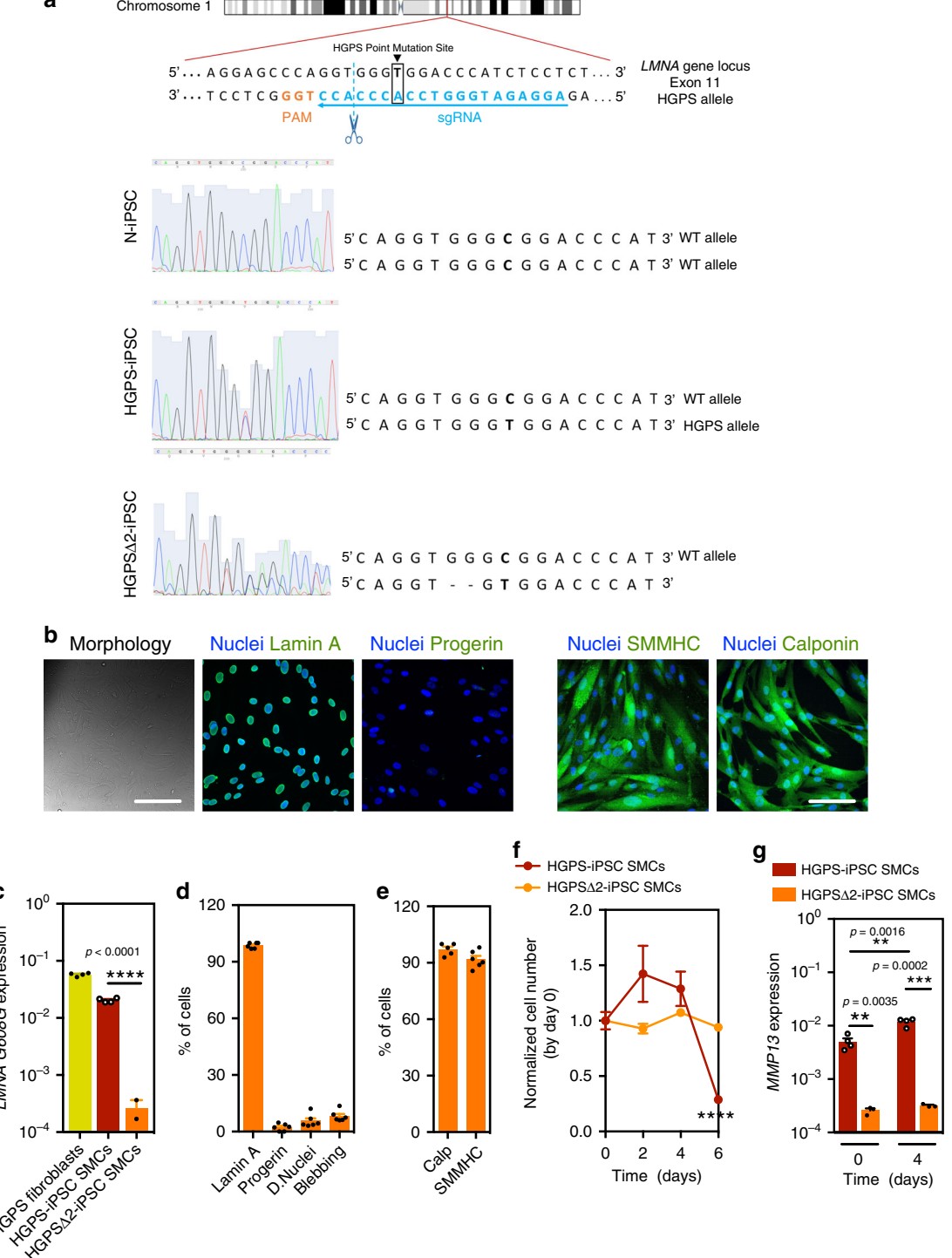

**MMP13 mediates HGPS-iPSC SMC loss under flow conditions.** Next, we tested whether the chemical inhibition of MMPs could prevent HGPS-iPSC SMC detachment. For this purpose, we used Batimastat (BB-94)[27], a broad spectrum matrix metalloprotease inhibitor (IC50 = 33 nM for MMP13[28]), and a specific MMP13 inhibitor pyrimidine-4,6-dicarboxylic acid, bis-(4-fluoro-3-methyl-benzylamide) (IC50 = 8 nM)[29]. Remarkably, both inhibitors significantly decreased the detachment of HGPS-iPSC SMCs cultured under arterial flow conditions (at least until day 12) (Fig. 4e), and this effect was much superior to that of lonafarnib (Supplementary Fig. 7d) or inhibition through the pyrophosphate calcification process[30] (Supplementary Fig. 11c). To confirm these results, HGPS-iPSC SMCs were subjected to siRNA knockdown of MMP13 and cultured under arterial flow conditions for 10 days (Fig. 4f, g). Our results show that the knockdown of MMP13 in SMCs increased the stability of HGPS-iPSC SMCs in flow culture conditions compared with non-treated cells. We also analyzed the effects of MMP13 and BB94 inhibition in HOZ mSMCs (Fig. 3e). Similar to what was observed with HGPS-iPSC SMCs, the detachment was significantly delayed when one of the inhibitors was used. To further demonstrate the importance of MMP13 in HGPS-iPSC SMC detachment, we enforced the expression of *MMP13* in somatic SMCs (hVSMCs) and cultured the modified cells in flow culture conditions

**Fig. 2 Expression of progeria and SMC markers in HGPSΔ2-iPSC SMCs. a** gRNA directs Cas9 nuclease against mutated exon 11 of *LMNA* gene, upstream the HGPS mutation, disrupting progerin, without altering lamin A and lamin C. Sanger sequencing for *LMNA* (NM_170707.4 transcript) exon 11 was performed for: N-iPSCs, HGPS-iPSCs and HGPSΔ2-iPSCs, confirming the deletion of two-base pairs in the HGPSΔ2-iPSCs. **b** Expression of lamin A, progerin, and SMC proteins monitored by immunofluorescence. Scale bar is 100 μm. $n = 6$ independent experiments. **c** Expression of *progerin* (*LMNA G608G* gene) in HGPS and HGPSΔ2 cell lines. Results are mean ± SEM ($n = 4$ technical replicates from a pool of three independent experiments). Statistical analyses were performed by a two-tailed unpaired Student's $t$ test. **d** Quantification of lamin A, progerin, dysmorphic nuclei, and nuclei blebbing. Results are mean ± SEM ($n = 6$ independent experiments). **** denotes statistical significance ($p < 0.0001$). **e** Percentage of cells that have been differentiated from HGPSΔ2-iPSCs that express SMC markers at protein level. Results are mean ± SEM ($n = 5$–6 independent experiments). **f** Number of cells per surface (mm$^2$) as quantified by high-content microscopy (at least three images (×10) have been quantified per time). The number of cells was evaluated after 6 days under arterial flow and was normalized by the number of cells present at day 0. $n > 3$ images examined over three independent experiments. **g** *MMP13* mRNA transcripts quantified by qRT-PCR analyses in HGPS-iPSC SMCs or HGPSΔ2-iPSC SMCs cultured under flow conditions. MMP13 mRNA transcripts were normalized by *GAPDH*. $n = 4$ technical replicates from a pool of three independent experiments. **, *** denote statistical significance ($p < 0.01$, $p < 0.001$). Statistical analyses were performed by a two-tailed unpaired Student's $t$ test.

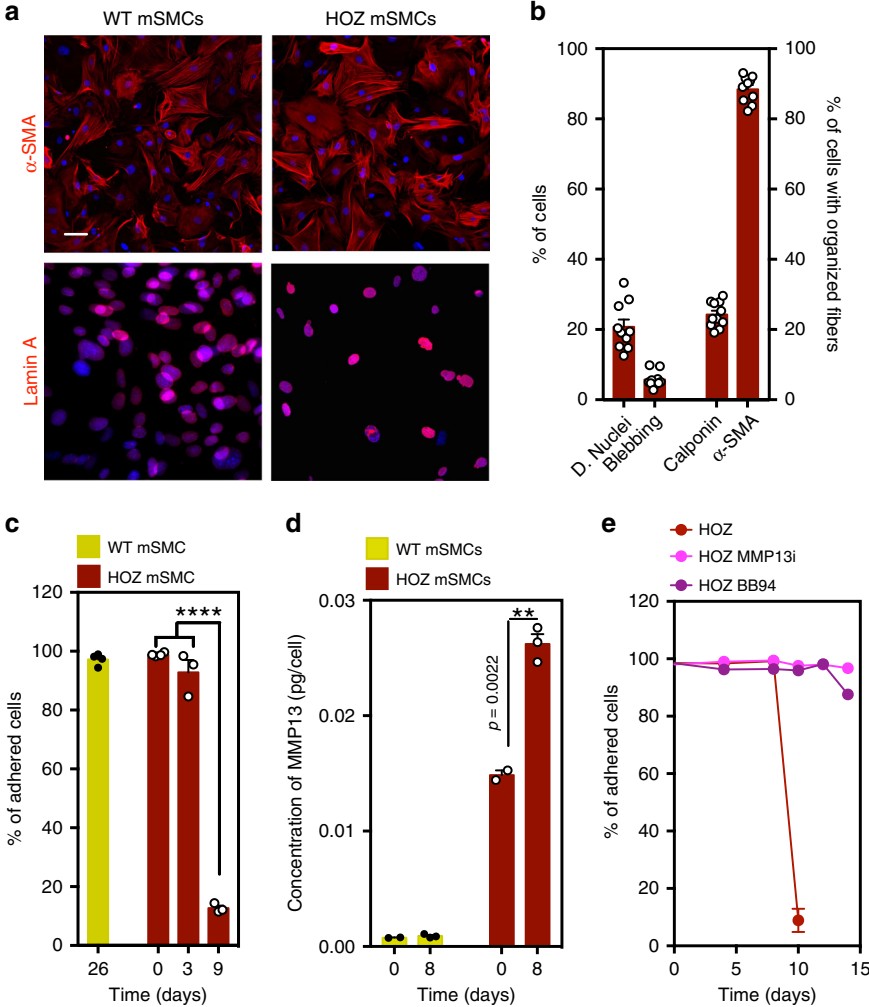

**Fig. 3 Characterization and impact of flow shear stress in SMCs isolated from wild-type (WT) and homozygous (HOZ) $Lmna^{G609G/G609G}$ mice. a** Mouse SMCs were cultured for 9–26 days in arterial flow conditions (120 dyne/cm$^2$). Immunofluorescence analyses performed on mouse SMCs (6-week-old WT and HOZ $Lmna^{G609G/G609G}$ mice) at passage 4 for α-SMA and Lamin A. Nuclei were stained with DAPI. Scale bar is 20 μm. $n = 3$–4 images examined over three independent experiments. **b** Percentage of dysmorphic nuclei, nuclei blebbing, and SMC organized fibers in mSMCs (assessed in static conditions). $n = 3$–4 images examined over three independent experiments. **c** Percentage of adhered cells over time. Cells were cultured under flow conditions. $n = 3$–4 independent experiments. Statistical analyses were performed by one-way ANOVA followed by Newman–Keuls's post test. **d** Quantification of MMP13 in HOZ mSMCs and WT mSMCs. Cells were analyzed at day 0 and day 8 under flow. Fluorescence signal was normalized by cell number. $n = 3$–4 independent experiments. Statistical analyses were performed by a two-tailed unpaired Student's $t$ test. **e** Percentage of adhered cells over time. Cells were cultured under flow conditions. $n = 5$–6 independent experiments. In graphs **b**–**e**, results are mean ± SEM. *,**,***,**** denote statistical significance ($p < 0.05$, $p < 0.01$, $p < 0.001$, $p < 0.0001$).

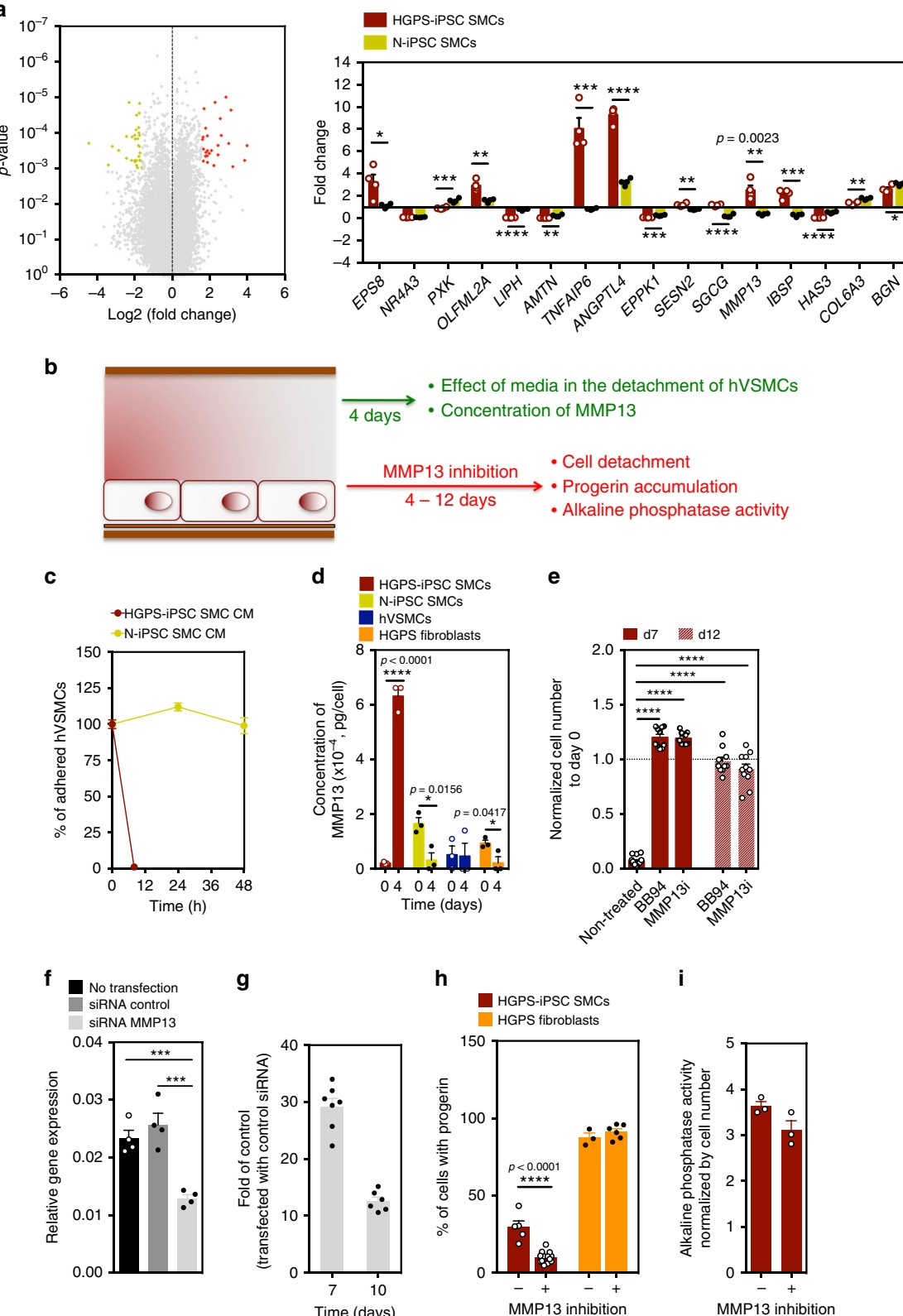

(Supplementary Fig. 19). Notably, the number of cells observed at day 7 is lower than the one observed in WT cells indicating that some of the modified cells were lost during the flow culture conditions.

We then asked whether the modulation of MMP13 activity could affect progerin expression associated with the vulnerability of HGPS-iPSC SMCs to flow shear stress. Interestingly, chemical inhibition of MMP13 in HGPS-iPSC SMCs cultured for 7 days in flow conditions reduced the percentage of progerin-positive cells (Fig. 4h); however, it did not decrease progerin expression in cells with high levels of progerin, such as HGPS fibroblasts. In addition, the chemical inhibition of MMP13 did not reduce the activity of alkaline phosphatase in HGPS-iPSC SMCs cultured for 7 days in flow conditions (Fig. 4i). Overall, the results obtained

**Fig. 4 MMP13 activity in HGPS-iPSC SMCs cultured under flow shear stress. a** Volcano plot representing differentially expressed genes in HGPS-iPSC-SMC cultured under flow conditions at day 0 and 4. Each point represents one of 53,617 genes. 26 and 31 genes were upregulated (red; fold change ≥ 3; $p < 0.001$) and downregulated (yellow; fold change ≤ 3; $p < 0.001$), respectively. Graph shows qRT-PCR validation for 16 genes with fold changes >3. Fold change was between days 0 and 4. Gene expression was normalized by the housekeeping gene *GAPDH*. Results are mean ± SEM, $n = 4$ technical replicates from a pool of three independent experiments. Statistical analyses were performed by a two-tailed unpaired Student's *t* test. **b** Schematic representation of the experimental protocol used. **c** Effect of HGPS-iPSC SMC or N-iPSC SMCs conditioned media (in both cases obtained after 4 days under flow conditions) on hVSMCs cultured under flow conditions. $n = 1$–5 images examined over three independent experiments. **d** Quantification of MMP13 activity (cell culture media) by ELISA. Cells were analyzed at days 0 and 4 under flow. Fluorescence signal was normalized by cell number. $n = 3$ independent experiments. Statistical analyses were performed by a two-tailed unpaired Student's *t* test. **e** Effect of MMP13 or BB94 inhibition in HGPS-iPSC SMC detachment. The number of cells was evaluated after 7 and 12 days under arterial flow and was normalized by the number of cells present at day 0. $n = 3$–5 images examined over three independent experiments. Statistical analyses were performed by one-way ANOVA followed by Newman–Keuls's post test. **f** MMP13 knockdown by siRNA in HGPS-iPSC SMCs. *MMP13* mRNA transcripts were quantified by qRT-PCR and normalized by *GAPDH*. Mean ± SEM ($n = 4$ technical replicates from a pool of three independent experiments). Statistical analyses were performed by one-way ANOVA followed by Newman–Keuls's post test. **g** Number of cells per microfluidic area during culture under flow shear conditions normalized by the number of cells in control experimental groups (i.e., cells transfected with control siRNA). $n = 7$ independent experiments for day 7 and $n = 6$ independent experiments for day 10. **h** Percentage of progerin-positive cells after 7 days under flow conditions with SmGM-2 media supplemented or not with MMP13 inhibitor. $n = 1$–5 images examined over three independent experiments. Statistical analyses were performed by a two-tailed unpaired Student's *t* test. **i** Activity of alkaline phosphatase in HGPS-iPSCs-SMC normalized by cell number per mm$^2$, in cells cultured 4 days under flow conditions. Cells were treated or not with MMP13 inhibitor. $n = 3$ independent experiments. In graphs **a**–**h**, results are mean ± SEM. *, **, ***, **** denote statistical significance ($p < 0.05$, $p < 0.01$, $p < 0.001$, $p < 0.0001$).

after chemical and genetic inhibition, the increase of MMP13 after flow shear stress and the effect of HGPS-iPSC SMC-conditioned media on cell detachment, indicate that MMP13 mediates SMC loss.

**Inhibition of MMP13 in *Lmna*$^{G609G/G609G}$ mice significantly increased the number of SMCs in aortic arch.** To confirm the importance of MMP13 dysregulation in progeroid animal models, we quantified MMP13 in the plasma of *Lmna*$^{G609G/G609G}$ and WT mice (Fig. 5a). The results showed that the levels of MMP13 were higher in mutant mice (Fig. 5b). Then, we asked whether the inhibition of MMP13 in *Lmna*$^{G609G/G609G}$ mice could decrease SMC loss. For this purpose, we generated double-mutant lines, *Lmna*$^{G609G/G609G}$*Mmp13*$^{-/-}$ and *Lmna*$^{G609G/G609G}$*Mmp13*$^{+/-}$ (Supplementary Fig. 20), and evaluated the heart rate and SMC loss in the aortic arch[20] of these mice at week 10 (Fig. 5a). Heart rate was chosen as a measure of the overall health status of the HGPS model and the derived double-mutant lines, given that bradycardia was a clinical abnormality evidenced in both *Lmna*$^{G609G/G609G}$ mouse as well as Zmpste 24$^{-/-}$ progeria mouse models[20,31]. Both double-mutant mice showed higher heart rates (Fig. 5d) and numbers of SMCs (Fig. 5c, e) in the aortic arch than *Lmna*$^{G609G/G609G}$ *Mmp13*$^{+/+}$ mice. Interestingly, *Lmna*$^{G609G/G609G}$ *Mmp13*$^{-/-}$ and *Lmna*$^{G609G/G609G}$*Mmp13*$^{+/-}$ mice showed a lower number of progerin-positive cells in the aortic arch than non-mutated mice (Fig. 5c, f). In addition, *Lmna*$^{G609G/G609G}$*Mmp13*$^{+/-}$ mice (but not *Lmna*$^{G609G/G609G}$*Mmp13*$^{-/-}$ mice) showed an increase of the aortic media thickness being similar to the non-mutated mice (Fig. 6a), as confirmed by orcein staining. We performed proteomic analyses of aortic arches from mutated and non-mutated mice ($n \geq 5$ mice per strain) using data independent acquisition mass spectrometry[32,33]. Principal component analysis based on 2260 proteins detected showed that the proteome profiles of aortic arches from *Lmna*$^{G609G/G609G}$*Mmp13*$^{+/-}$ mice were more closely related to the profile of WT mice to that of *Lmna*$^{G609G/G609G}$ *Mmp13*$^{+/+}$ mice (Fig. 6c). From the 161 proteins differentially expressed between the mutant and WT mice aortic arches ($q < 0.05$ and abs(log$_2$ fold change) > 0.58), ~25% of the proteins had similar expression in *Lmna*$^{G609G/G609G}$*Mmp13*$^{+/-}$ mice and WT mice (Fig. 6c and Supplementary Data 4).

Motivated by these results, we then tested a therapeutic approach to reduce SMC loss in *Lmna*$^{G609G/G609G}$ *Mmp13*$^{+/+}$ mice. For this purpose, we used Batimastat because human safety has been previously demonstrated in clinical trials[34].

*Lmna*$^{G609G/G609G}$ *Mmp13*$^{+/+}$ mice at week 5 were intraperitoneal (IP) injected five times a week (Fig. 7a). At week 10, Batimastat-treated *Lmna*$^{G609G/G609G}$ *Mmp13*$^{+/+}$ mice had similar heart rates to non-treated animals (Fig. 7c); however, they showed higher SMCs in the aortic arch than non-treated mice, as confirmed by cell nuclei counts and verified by the increase levels of SMC markers determined by qRT-PCR analyses (Fig. 7b, d, e). No differences were observed between non-treated and Batimastat-treated mice regarding progerin accumulation in the aortic arch (Supplementary Figs. 20c). Overall, our data shows that the in vivo inhibition of MMP13 by genetic or chemical interventions yielded mice having significantly higher numbers of SMCs in the aortic arch.

**Activation of MMP13 is mediated by the activation of the glycocalyx.** The glycocalyx is a surface layer of proteoglycans and glycosaminoglycans that are immobilized in the cell membrane. Glycocalyx components have been shown to be involved in flow shear stress sensing by SMCs[35,36]. To identify the mechanism underlying the upregulation of MMP13 in HGPS-iPSC SMCs cultured under arterial flow, we analyzed glycocalyx gene mRNA transcripts (Fig. 8b). Interestingly, glycocalyx transcripts were upregulated in HGPS-iPSC SMCs cultured under flow conditions for 4 days (Fig. 8b). From these upregulated genes, syndecan 2 gene (*SDC2*), which encodes the transmembrane (type I) heparan sulfate proteoglycan, was also upregulated in hVSMCs or N-iPSC SMCs cultured for 4 days in flow conditions (Supplementary Fig. 21). Because not all the glycocalyx mRNA transcripts were upregulated in hVSMCs and N-iPSC SMCs, the results suggest that the composition of glycocalyx is likely different in these cells when compared with HGPS-iPSC SMCs. Next, we analyzed the expression of heparan sulfate at the protein level. In contrast to control cells, the expression of heparan sulfate increased when HGPS-iPSC SMCs were cultured under flow conditions (Fig. 8a). Importantly, the enzymatic cleavage of heparan sulfate by heparinase III (Supplementary Fig. 22) decreased MMP13 concentration in the cell culture media (Fig. 8c) and significantly decreased the detachment of HGPS-iPSC SMCs cultured under flow conditions (Fig. 8d). Moreover, the enzymatic cleavage of heparan sulfate slightly decreased alkaline phosphatase activity (Fig. 8e).

To further investigate a potential ECM target of MMP13 in SMCs, we monitored the expression of ECM components in hVSMCs, HUAECs, N-iPSC SMCs, and HGPS-iPSC SMCs. Our

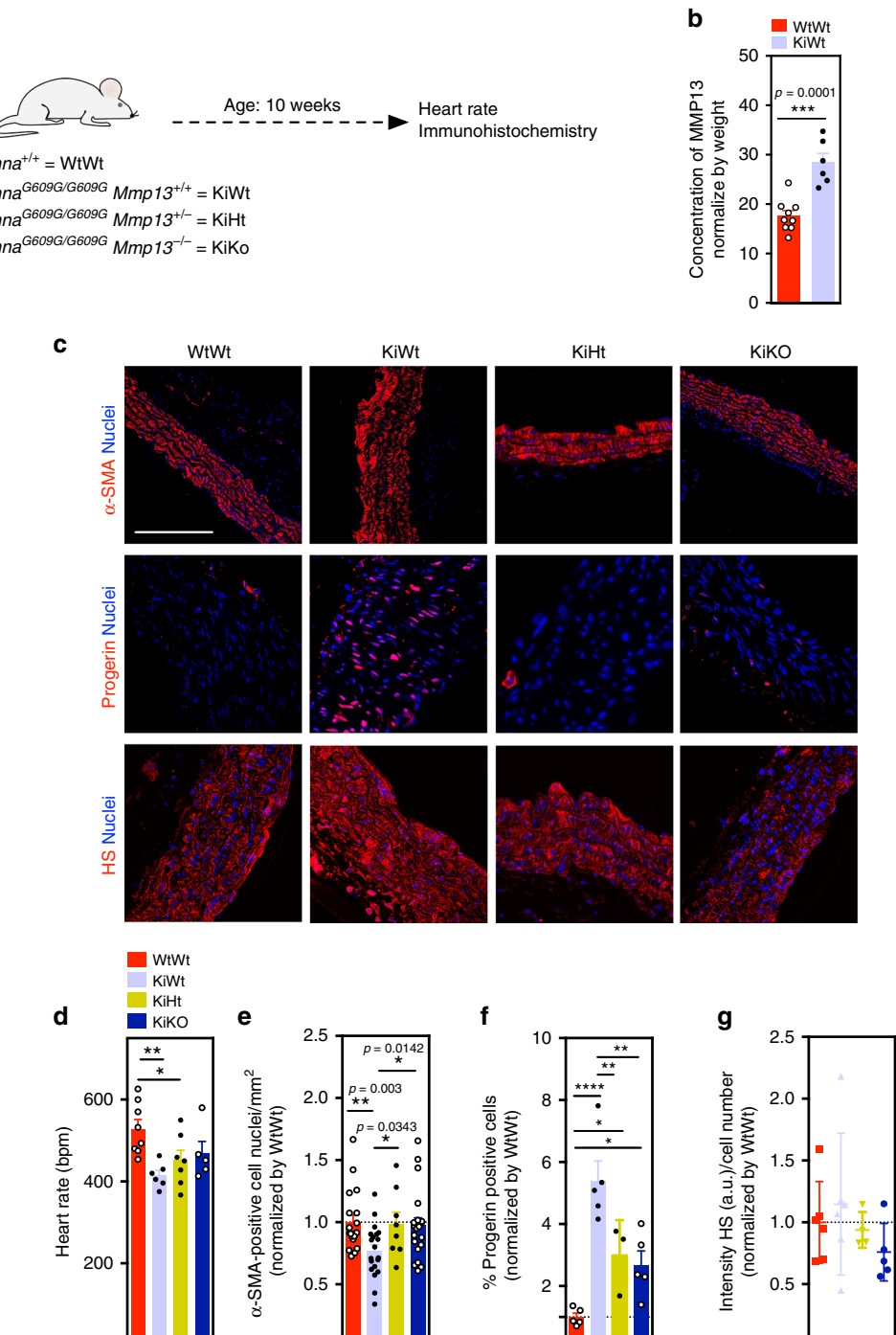

**Fig. 5 MMP13 inhibition significantly increases SMC number in aortic arch of *Lmna*^G609G/G609G mice. a** Schematic representation of the animal protocol. WtWt, KiWt, KiHt, and KiKo mice (age: 10 weeks) were evaluated. **b** Quantification of MMP13 activity (plasma from WtWt, $n = 9$, and KiWt, $n = 6$, mice) by ELISA. Fluorescence signal was normalized by mice weight. Statistical analyses were performed by a two-tailed unpaired Student's *t* test. **c** Immunofluorescence analyses in the aortic arch for α-SMA, progerin, and heparan sulfate (HS). Cell nuclei were stained with DAPI. Scale bar is 100 μm for α-SMA staining and 50 μm for progerin and heparan sulfate staining. For α-SMA staining, $n = 5$ animals, except for KiHt (four animals). For progerin staining, $n = 5$ animals, except for KiHt (three animals). For heparan sulfate $n = 6$ WtWt, $n = 6$ KiWt, $n = 4$ KiHt, and $n = 5$ for KiKo. **d** Heart rates in mice ($n = 8$ WtWt, $n = 6$ KiWt, $n = 7$ KiHt, and $n = 5$ KiKo). Statistical analyses were performed by one-way ANOVA followed by Newman–Keuls's post test. **e** Number of SMC nuclei in aortic arch per tissue area (mm²) ($n = 2$–3 slides examined over five animals, except for KiHt (four animals)). Statistical analyses were performed by a two-tailed unpaired Student's *t* test. **f** Percentage of progerin-positive cells in SMCs. $n = 5$ animals, except for KiHt (three animals). Statistical analyses were performed by one-way ANOVA followed by Newman–Keuls's post test. **g** Expression of heparan sulfate as evaluated by immunofluorescence. Intensity of heparan sulfate was calculated in each picture (at least 16 pictures per condition) and normalized by cell number mice ($n = 6$ WtWt, $n = 6$ KiWt, $n = 4$, KiHt and $n = 5$ KiKo). In **b**, **d**–**g**, results are mean ± SEM. \*, \*\*, \*\*\*, \*\*\*\* denote statistical significance ($p < 0.05$, $p < 0.01$, $p < 0.001$, $p < 0.0001$).

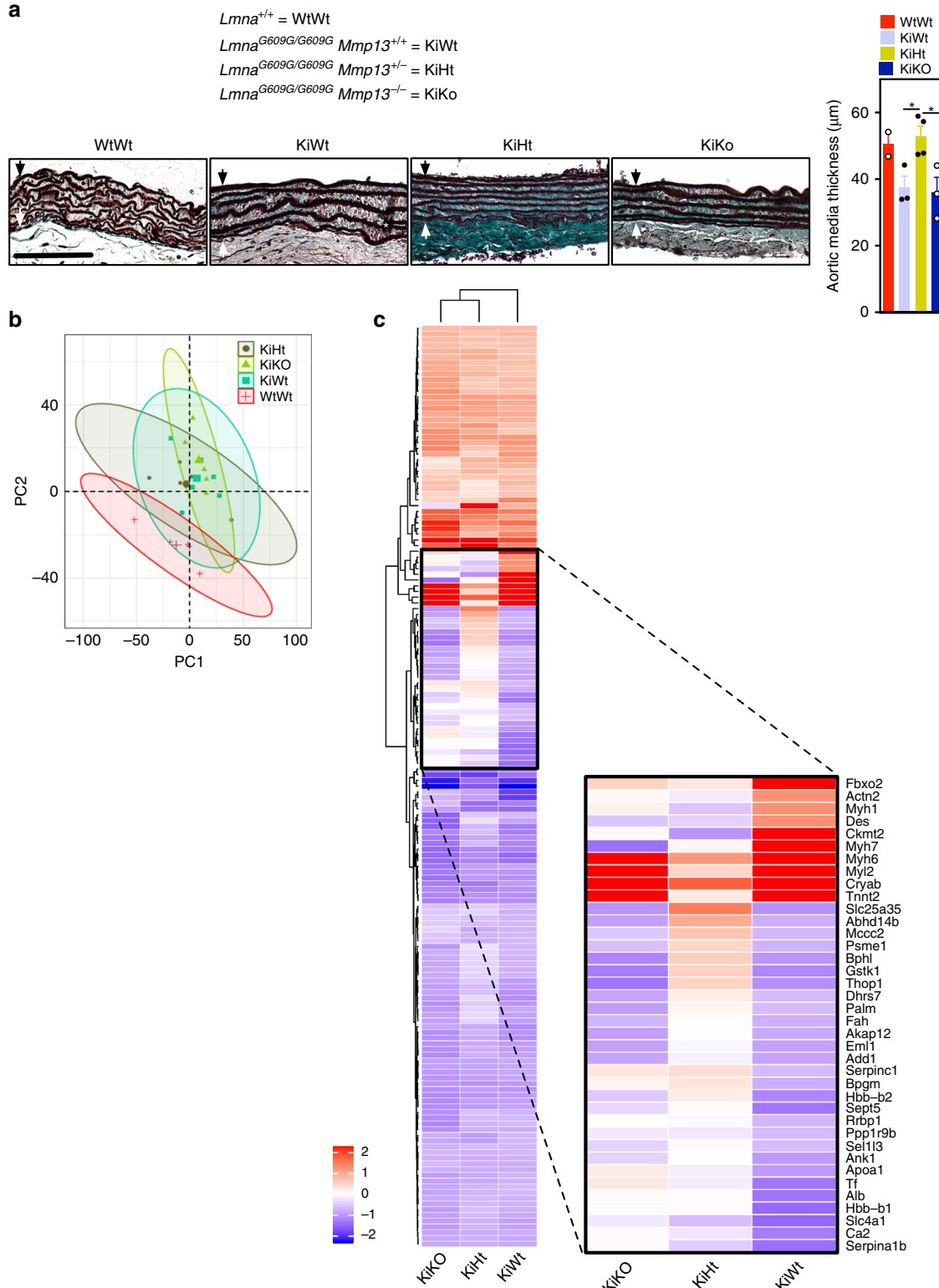

results indicate that hVSMCs express higher levels of mRNA that encode collagen 1A1, collagen 3A1, collagen 4A2, and collagen 6A3 than HUAECs (Supplementary Fig. 21c). It has been shown that MMP13 degrades very efficiently the native helix of all fibrillary collagens, including collagen type I[37]. Our proteomic results indicate that indeed collagen 1A1 is upregulated in HGPS-iPSC SMCs exposed to flow conditions (Supplementary Fig. 16b) and thus it may be a potential target for MMP13. Overall, our

results indicate that activation of MMP13 is mediated, at least in part, by glycocalyx activation.

## Discussion

In this study, we developed a microfluidic chip formed by a monoculture or a co-culture of HGPS-SMCs (generated from iPSCs) with ECs to study the reason underlying HGPS-SMC

**Fig. 6 Proteins differentially expressed in the aortic arch at week 10 on wild-type and mutant (*Lmna*$^{G609G/G609G}$*Mmp13*$^{-/-}$ and *Lmna*$^{G609G/}$**
**$^{G609G}$*Mmp13*$^{+/-}$) mice. a** Orcein-stained ascending aorta (elastic fibers stain in dark brown/black). Black arrow defines the internal elastic lamina while the white arrow defines the adventitial border. Images illustrate morphological changes rather than aortic media thickness differences. KiWT mice show less compact elastic lamellae and higher irregular profiles of the elastic lamellae (labeled with *) than the other mice. Scale bar is 50 μm. In graph, aortic media thickness was measured from the internal elastic lamina to the adventitial border. Black arrow defines the internal elastic lamina while the white arrow defines the adventitial border. Results are mean ± SEM, $n = 3$ animals, except for KiHt (four animals). * denotes statistical significance ($p < 0.05$). Statistical analyses were performed by one-way ANOVA followed by Newman–Keuls's post test. **b** Principal component analysis (PCA) of proteome profiles obtained from aortic arches of wild-type (WtWt) and mutant (KiWt, KiHt, KiKo) mice. **c** Heatmap based on 161 protein groups differentially expressed between KiWt and WtWt mice, in aortic arch, at week 10 ($q < 0.05$ and abs(log$_2$ fold change) > 0.58). Progerin is a mutated protein and thus not identified by the mass spectrometry. MMP13 is a secreted protein and the levels in cells were not detectable by mass spectrometry. For comparison purposes, the protein fold changes of WtWt vs. KiHt and WtWt vs. KIKo were included in the heatmap. Blue color indicates proteins downregulated in KiWt, KiHt, or KiKo as compared with WtWt, whereas red color corresponds to proteins upregulated in KiWt, KiHt, or KiKo as compared with WtWt. $n = 6$ for KiWt and $n = 5$ for WtWt, KiHt, and KiKo; age: 10 weeks.

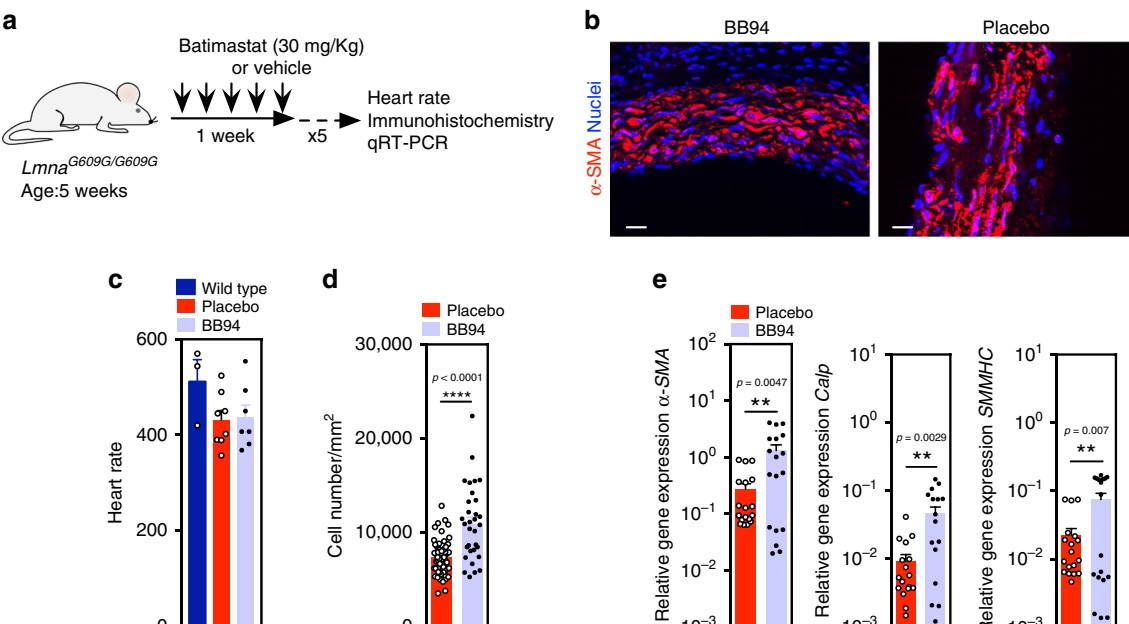

**Fig. 7 MMP treatment using BB94 significantly increases SMC number in aortic arch of *Lmna*$^{G609G/G609G}$ mice. a** Schematic representation of the animal protocol. *Lmna*$^{G609G/G609G}$ mice ($n = 8$ for treatment group and control group; age: 5 weeks) were IP injected five times a week (30 mg/kg/day; 3 mg/mL in PBS). **b** Immunofluorescence analyses performed on mouse SMC for α-SMA showing higher number of SMCs in treated aortic arch. Cell nuclei were stained with DAPI. SMCs were stained for α-SMA. Scale bar is 100 μm. For BB94 treatment $n = 5$ animals. For placebo treatment $n = 7$ animals. **c** Heart rates in mice. Wild-type mice were not exposed to BB94. $n = 3$ for wild-type mice, $n = 8$ for placebo treatment group and $n = 7$ for BB94 treatment group. **d** Number of SMC nuclei in aortic arch per tissue area (mm$^2$) in mice treated or not with BB94. For BB94 treatment, $n > 6$ images examined over five animals. For placebo treatment, $n > 9$ images examined over seven animals. **e** Expression of SMC genes in aortic arches of mice treated or not with BB94. Gene expression was normalized by the housekeeping gene *GAPDH*. $n > 3$ technical replicates over six animals. **, ***, **** denote statistical significance ($p < 0.01$, $p < 0.001$, $p < 0.0001$). Statistical analyses were performed by a two-tailed unpaired Student's *t* test **d** and **e**.

vulnerability to flow shear stress. To generate the chip, we (i) developed a protocol to differentiate HGPS-iPSCs into functional HGPS-SMCs, (ii) demonstrated that HGPS-iPSC SMCs shared similar properties with other known progerin-expressing cells, (iii) confirmed that HGPS-iPSC SMCs were vulnerable to arterial flow shear stress, and (iv) validated the results in ex vivo SMCs isolated from *Lmna*$^{G609G/G609G}$ mice. Using the chip, we have identified MMP13 upregulation as an important mediator of HGPS-SMC vulnerability to flow shear stress and we confirmed MMP13's role in vivo in *Lmna*$^{G609G/G609G}$ mice (Fig. 8f). MMP13 is upregulated in a number of pathological states including atherosclerosis and rheumatoid arthritis[38]. The upregulation of MMP13 in HGPS-SMCs cultured under arterial flow conditions is in line with examples in the literature showing that enzymatic ECM remodeling is significantly altered in HGPS cells[39–41].

Multiple protocols have been described in the literature for the differentiation of iPSCs into SMCs, either via an intermediate progenitor stage or directed differentiation[14,42–44]. These protocols are highly variable in terms of SMC differentiation efficiency, timescale, and functionality (nondividing contractile phenotype vs. proliferative phenotype, secretory profile), likely due to the choice of precursor population to derive the SMC subtypes, the chemical composition of the differentiation medium, as well as the choice of inductive SMC factors (e.g., PDGF-BB, TGF-β1, retinoic acid). Three previous studies have reported the differentiation of HGPS-iPSCs into SMCs[7,9,45] by direct differentiation[7] or by using an intermediate progenitor (i.e., mesenchymal stem cells[45] or CD34$^+$ cells[9]). In some cases, SMCs were not terminally differentiated (as confirmed by the expression of SMMHC)[7], in others the percentage of SMCs was relatively low

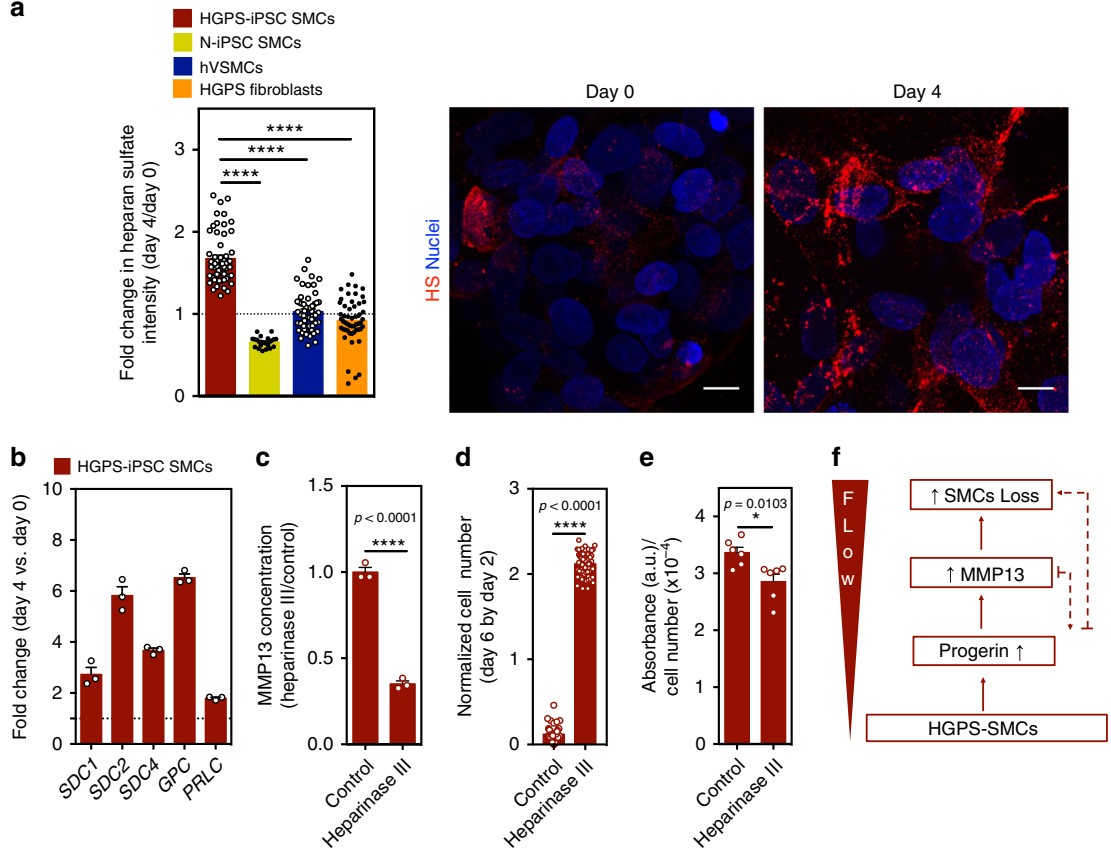

**Fig. 8 MMP13 expression in SMCs is triggered by an increase in heparan sulfate. a** Cells were cultured under flow conditions for 4 days and the expression of heparan sulfate was evaluated by immunofluorescence. Intensity of heparan sulfate was calculated in each picture and normalized by cell number. The normalized fluorescence intensity at day 4 was divided with the one at day 0. Scale bar is 10 μm. $n > 4$ images examined over six independent experiments. Statistical analyses were performed by one-way ANOVA followed by Newman–Keuls's post test. **b** Gene expression of glycocalyx markers (*SDC1*: syndecan 1, *SDC2*: syndecan 2, *SDC4*: syndecan 4, *GPC*: glypican, *PLC*: perlecan), as evaluated by qRT-PCR, in HGPS-iPSC SMCs cultured under flow conditions. Gene expression was normalized by the housekeeping gene *GAPDH*, and the normalized gene expression at day 4 divided by day 0. $n = 3$ technical replicates from a pool of three independent experiments. **c** HGPS-iPSCs-SMC cultured under flow condition were treated or not with heparinase III and the number of cells per microfluidic area during culture was calculated and normalized by the number of cells present at day 2. $n = 3$ independent experiments. Statistical analyses were performed by a two-tailed unpaired Student's *t* test. **d** Quantification of MMP13 activity (cell culture media) by ELISA. Cells were analyzed at day 4 under flow. Fluorescence signal was normalized by cell number and then by control experimental group. $n > 9$ images examined over six independent experiments. Statistical analyses were performed by a two-tailed unpaired Student's *t* test. **e** Expression of alkaline phosphatase in HGPS-iPSCs SMC, normalized by cell number per mm², in cells cultured 4 days under flow conditions. Cells were treated or not with heparinase III. $n = 2$ technical replicates over three independent experiments. Statistical analyses were performed by a two-tailed unpaired Student's *t* test. In **a–e**, results are mean ± SEM. *, **, ***, **** denote statistical significance ($p < 0.05$, $p < 0.01$, $p < 0.001$, $p < 0.0001$). **f** Summary of the results.

(i.e., only 50–60% of the differentiated cells showed specific SMC markers including α-SMA, calponin, and SMMHC)[45] and no indication of SMC functionality[9] (e.g., contractility, intracellular accumulation of calcium after exposure to vasoactive agents) was reported. In the present study, we showed that the differentiation of HGPS-iPSCs induces the activation of the NOTCH signaling pathway, a hallmark of progerin-expressing cells[15]. This is observed in the CD34+ progenitor cells and after their differentiation into SMCs. The CD34+ cells have been reported to express KDR and CD31[43] and, thus, are likely of lateral plate mesoderm origin[42,44]. Importantly, the differentiated cells express high levels of all the SMC markers analyzed (α-SMA, calponin and SMMHC), are contractile in response to the muscarinic receptor agonist, carbachol, as observed in typical human aortic SMCs, and, when matured in culture for ~30 days, they express progerin. Therefore, our differentiation protocol compares favorably to other protocols in term of SMC yield and functionality. Interestingly, HGPS-iPSC SMCs express lower levels of calponin than in N-iPSC SMCs but the reason and possible

implications behind this phenotypic difference remain to be determined. Nevertheless, most of the HGPS-iPSC SMCs expressed calponin at the protein level, both at the induction and maturation steps (Supplementary Figs. 3, 4). A previous study has reported heterogeneous sized calponin 1-staining inclusion bodies in the cytoplasm of HGPS-SMCs[9]; however, such structures were not observed in the current study.

It has been reported that in WT animals the aorta was one of the tissues with the highest expression of lamin A, while in progeroid animals the aorta was the first place where progerin was detected[8]. This explains the highest susceptibility of HGPS-SMCs located in the aorta to biomechanical forces. It has been reported that mouse SMCs overexpressing progerin exposed to biomechanical forces detach from the culture vessel after substrate stretching and die[8]. Yet, the mechanism of SMC detachment is still poorly understood. Our study indicates that MMP13 mediates SMC detachment as chemical or genetic inhibition of MMP13 reduces significantly SMC loss. In addition, we found that the accumulation of progerin is a mediator and not the cause

of SMC detachment because HGPS fibroblasts accumulate high levels of progerin and do not detach in flow conditions. Yet, both inhibition of progerin by morpholinos and the knockout of the HGPS mutant allele in HGPS-SMCs decreased or prevented SMC detachment in flow culture conditions.

Although $Lmna^{G609G/G609G}Mmp13^{+/-}$ and $Lmna^{G609G/G609G}$ $Mmp13^{-/-}$ mice showed similar amelioration of SMCs loss in the aortic arch, our proteomic analyses in the same tissue showed that $Lmna^{G609G/G609G}Mmp13^{+/-}$ mice had a closer protein profile to WT than $Lmna^{G609G/G609G}Mmp13^{-/-}$ mice. This was consistent with the media thickness size, which was more similar between WT mice and $Lmna^{G609G/G609G}Mmp13^{+/-}$ than to $Lmna^{G609G/G609G}$ $Mmp13^{+/+}$ mice. Previous studies have shown that $Mmp13^{-/-}$ mice had defects in vascularization[46] and thus the full deficiency of MMP13 in the aortic arch might not be desirable to establish a phenotype closer to the normality.

The accumulation of proteoglycans in Progeria mouse models[47] as well as in atherosclerotic lesions in HGPS individuals[5] has been demonstrated. According to our results, the upregulation of MMP13 in HGPS-SMCs under flow conditions is mediated by the upregulation of glycocalyx components, which have been previously implicated as flow shear stress sensors[35]. The inhibition of components of glycocalyx by enzymatic treatment decreases significantly the MMP13 levels, the osteogenic program of SMCs and SMCs detachment. Although the connection between MMP13 and glycocalyx has been shown previously for non-disease SMCs, we show here that the accumulation of glycocalyx is responsible for the MMP13 expression under shear stress conditions, which subsequently leads to the loss of HGPS-SMCs. It is possible that the activation of MMP13 expression triggered by an upregulation of glycocalyx is mediated by the phosphorylation of ERK and FAK and the activation of c-Jun signaling pathway[35] or mediated via NOTCH signaling pathway[48]. Our in vivo results indicated that the expression of heparan sulfate proteoglycans in the aortic arches at week 10 on $Lmna^{G609G/G609G}Mmp13^{+/+}$ mice was not statistically different from the expression profile found in WT mice. It is possible that further time is needed to see this upregulation as seen in other progeroid animal models[3,16] or in HGPS individuals[5]. Since the upregulation of heparan sulfate was not observed in $Lmna^{G609G/G609G}Mmp13^{+/+}$ mice, it is not surprising that we could not observe a statistical decrease in heparan sulfate in $Lmna^{G609G/G609G}Mmp13^{-/-}$ mice.

The in vivo treatment results presented here using the MMP inhibitor Batimastat open possibilities for the treatment of HGPS and vascular aging[49,50]. Batimastat acts as an inhibitor of metalloproteinase activity by binding the zinc ion in the active site of MMPs. Batimastat has been used previously for the treatment of human cancer (e.g., malignant ascites[51] and malignant pleural effusions[34]) with demonstrated results and few side-effects in phase I/II clinical trials. Therefore, the current study proposes Batimastat as a drug to be considered for future Progeria trials. It should be noted that most of the compounds identified so far in preclinical tests to treat Progeria have been focused: (i) in the reduction of progerin quantities, by either reducing its production or increasing its degradation; (ii) in the reduction of progerin toxicity by targeting its aberrant prenylation: or (iii) in the identification of compounds capable of restoring pathological phenotypes downstream of progerin accumulation. Although these treatments showed encouraging results in preclinical studies and, in some cases in clinical trials, they do not address SMC loss over time. The administration of a drug that prevents SMC loss in early stages of disease combined with drugs that further reduce accumulation of progerin and progerin toxicity could be of added value to extend the lives of HGPS individuals.

Future studies should address the effect of SMC preservation in large vessels in the lifespan of the animals. It is possible that the prevention of SMC loss from the large arteries might be insufficient to lead to a significant increase in animal lifespan. Evidence collected at week 12[8] (before the $Lmna^{G609G/G609G}$ died of progeria disease) in a therapy that ameliorated SMC loss showed no significant alterations in terms of body weight (which is correlated with lifespan[20]). Our study performed for 10 weeks showed also no significant changes in body weight (Supplementary Fig. 20d) between $Lmna^{G609G/G609G}Mmp13^{+/-}$ mice and $Lmna^{G609G/G609G}Mmp13^{+/+}$ mice. Therefore, it is possible that therapies which ameliorate SMC loss should be combined with therapies that further reduce the level of progerin in cells of the major organs, in particular the heart, which seems to present electrical defects[31]. Another issue that deserves further investigation is the relationship between MMP13 and progerin. Both in vitro and in vivo results indicate that the silencing of MMP13 leads to a significant reduction of progerin in SMCs and the reason for this pattern is presently not known. Overall, our study demonstrates that the control of MMP13 expression decreases the vulnerability of SMCs in large vessels and this strategy may be of potential value to reduce the impact of the disease in Progeria individuals.

## Methods

**iPSCs culture and differentiation.** iPSCs were generated from HGPS skin fibroblasts provided by Coriell Institute and characterized according to Nissan et al.[10]. iPSCs were derived using the Yamanaka's original method with OCT4, KLF4, SOX2, c-MYC, transferred using retroviral vectors. All HGPS cells were obtained from Coriell Institute for Medical Research, which in turn were collected under Institutional Review Board approval and individual informed consent (https://www.coriell.org/0/Sections/Support/NIA/Model.aspx?PgId=351). HGPS-iPSCs clone 1 (passages 43-51); HGPS-iPSCs clone 2 (passages 35-42), and N-iPSCs (passages 30-35) were maintained on mitotically inactivated mouse embryonic fibroblast (MEF) feeder layer, according to Ferreira et al.[43]. Culture medium for the present work consisted of 80% KO-DMEM (Life Technologies), 0.5% L-glutamine (Life Technologies), 0.2% β-mercaptoethanol (Sigma), 1% nonessential amino acids (Invitrogen), and penicillin-streptomycin (50 U/mL:50 mg/mL) (Lonza), supplemented with 20% KnockOut™ Serum Replacement (Gibco®) and 10 ng/mL of b-FGF (Peprotech). Colonies were expanded by routine passage every 3/4 days with 1-mg/ml collagenase type IV (Life Technologies). To induce EBs formation, the iPSCs were treated with collagenase IV (1 mg/mL, Gibco) for 1 h and then transferred (2:1) to low attachment plates (Corning) containing 10 mL of differentiation medium (80% KO-DMEM (Life Technologies), 20% fetal bovine serum (FBS, Invitrogen), 0.5% L-glutamine (Life Technologies), 0.2% β-mercaptoethanol (Sigma), 1% nonessential amino acids (Invitrogen), and penicillin-streptomycin (50 U/mL:50 mg/mL) (Lonza). EBs were cultured for 10 days at 37 °C, 5% $CO_2$ in a humidified atmosphere, with media changes every 2 days. $CD34^+$ cells were isolated from EBs at day 10 using MACS (Miltenyi Biotec). The percentage of $CD34^+$ cells in EBs was between 0.4 and 1.5%. Isolated cells were grown on 24-well plates ($\sim3 \times 10^4$ cells/cm$^2$) coated with 0.1% gelatin in the presence of EGM-2 (Lonza) supplemented with $PDGF_{BB}$ (50 ng/mL, Peprotech). After four passages, the medium was replaced by Smooth Muscle Growth Medium-2 (SmGM-2) (Lonza CC-3182) (maturation medium), for additional four passages. hVSMCs (Lonza) were used as controls for the differentiation studies. Cell cultures were maintained at 37 °C, 5% $CO_2$ in a humidified atmosphere, with media changed every 2 days. A step-by-step protocol can be found at Protocol Exchange[52].

**Cell culture under arterial flow conditions.** A suspension of HGPS-iPSC SMCs (clone 1), HGPS-iPSC SMCs (clone 2), N-iPSC SMCs, hVSMCs, or HGPS fibroblasts between $5 \times 10^4$ and $1.3 \times 10^5$ cells/cm$^2$ was applied to the entry port of an IBIDI channel (μ-Slide I $^{0,4}$ Luer, or μ-Slide VI $^{0,4}$ Luer, IBIDI) and allowed to flow inside by capillary force. After 4 h, a confluent cell layer was formed, which was then perfused with SmGM-2 medium or fibroblasts medium (DMEM supplemented with FBS (20%, v/v, Gibco), sodium pyruvate (Sigma, 1 mM) and penicillin-streptomycin (50 U/mL:50 mg/mL)) at physiological flow rate (20 dyne/cm$^2$). Unless specified, all tests were performed at days 0 and 4 on flow culture conditions. Cell number and cell clumps were determined on slides stained with DAPI (20×) and normalized by image area (0.3524 mm$^2$). Cell clumps areas were evaluated by ImageJ software.

**MMP activity.** MMP activity was quantified on cell extracts by a fluorometric red assay kit (Abcam). Cell extracts were obtained by incubating the cells with Triton X-100 (0.5%, v/v, in PBS, Sigma) for ~15 min, the cells were centrifuged

and the supernatant collected. Part of cell extract (25 μL) was added to 4-aminophenylmercuric acetate (25 μL, 2 mM) and incubated for 40 min at 37 °C. Then, a MMP red substrate (50 μL) was added to the mixture and the fluorescence intensity measured in a fluorimeter (Ex/Em = 540/590 nm) after 1 h, at room temperature. An ELISA kit was used to quantify the expression of MMP13 protein. Cell culture media collected from different experiments and plasma from WtWt and KiWt mice was used for MMP13 quantification (MMP13 human ELISA kit from Abcam and Mmp13 mouse ELISA kit from USCN) according to manufacture recommendation. Briefly, standard or sample (100 μL) was added to each well and incubated for 1 h at 37 °C. Then, solutions were aspirated and detection reagent A (100 μL) was added and incubated for 1 h at 37 °C. After washing three times, detection reagent B (100 μL) was added, incubated 30 min at 37 °C and washed five times. Substrate solution (90 μL) was then added and left to incubate for 10–20 min at 37 °C. Finally, stop solution was added to the wells (50 μL) and the absorbance of the solution monitored at 450 nm.

**Glycocalyx analyses**. To quantify the intensity of heparan sulfate, cells were stained with heparan sulfate (1:50 for staining, 10E4 Epitope, USBiological) as described in supplementary information. ImageJ software was used to quantify the overall intensity of each image, which was then normalized for cell number. Heparinase III from Flavobacterium heparinum (Sigma), was used for the enzymatical degradation of heparan sulfate. Briefly, HGPS-iPSC-SMCs cultured under flow condition during 4 days were subjected to heparinase III treatment (0.5 U/ml for 30 min at 37 °C), and the number of cells per microfluidic area during culture was calculated.

**Treatment of *Lmna*^G609G/G609G^ mice with Batimastat**. Sixteen *Lmna*^G609G/G609G^ mice (male and female) were used. After sex and body weight randomization, animals were allocated in different groups and treated with vehicle (eight *Lmna*^G609G/G609G^ control mice) or BB94 inhibitor (eight *Lmna*^G609G/G609G^ mice treated with Batimastat in vehicle solution). IP injections were used to administrate 30 mg/kg/day of BB94 at 3 mg/mL in PBS containing 0.01% Tween 80. The treatment was administered five times per week during 6 weeks (from week 5 to week 10). The treatment duration was reduced from 10 to 6 weeks due to intra-abdominal accumulation of BB94 (precipitate). At the end of week 10 the mice were sacrificed, and the selected parameters were evaluated.

**Double mutant generation and heart rate monitoring**. *Lmna*^G609G/G609G^ mice present infertility as described by Osorio and colleagues, therefore the *Lmna*^G609G/G609G^ *Mmp13*^−/−^ mice (KiKO) were generated from *Lmna*^G609G/+^ and *Mmp13*^−/+^ heterozygous (in a C57BL/6 background) as our colony founders (F0). The offspring presenting the *Lmna*^G609G/+^ *Mmp13*^−/+^ (F1) were used for further back-crossing to generate the Progeria double mutants (KiKO) and Progeria control (KiWT) genotypes used in the present study. All mice were bred in-house in ventilated cages in a temperature and humidity-controlled room with a 12-h light/dark cycle. The founder *Lmna*^G609G/+^ mice were a kind gift from Dr Lopez-Otin[20].

Genotyping analyses were performed to select those mice carrying the *Lmna*^G609G^ mutation in homozygosis and the MMP13 deficiency or WT genes. Briefly, DNA was obtained from tails using the PureLink® Genomic DNA Mini Kit (Invitrogen) and DNA yields used for the PCR reaction using the Platinum®Taq DNA Polymerase (Invitrogen) and a combination of custom-designed oligonucleotides for the amplification of the *Lmna* and *Mmp13* genes. PCR products were run in agarose gels with RedSafe Nucleic Acid Staining Solution (Labotaq) for detecting the amplified Lmna DNA fragments (G609G allele at 240 bp and WT at 100 bp) and Mmp13 fragments (KO at 1485 bp and WT at 1300 bp).

For heart rate monitoring mice were anesthetized with isoflurane (5% induction and 2% maintenance in oxygen) and a mouse paw pulse sensor (Kent Scientific Corporation) placed in the hindlimb paws until stable heart beats were detected and recorded by the PhysioSuite™ noninvasive monitoring system (Kent Scientific Corporation). During the procedure and until mice recovered from anesthesia body temperature was controlled with a heating pad.

All procedures were approved by the Ethics Committee of Animal Experimentation (CCEA 57/16) of the Vall d'Hebron Research Institute and were conducted in compliance with Spanish legislation and in accordance with the Directives of the European Union.

**Proteomic analysis of aortic arches from WT and mutant mice**. Formalin-fixed and paraffin-embedded slices of aortic arch (4 um) were processed for mass spectrometry analysis as described in the Supplementary Material Information and according to Heinze et al.[33]. The obtained peptides were analyzed using Data Independent Acquisition[53] on an Orbitrap Fusion Lumos mass spectrometer (Thermo Fisher) connected online with a Waters nanoAcquity UPLC system (details regarding instrument settings and data acquisition parameters can be found in the Supplementary Information). Spectral library generation, data processing, and differential expression analysis were performed in Spectronaut 11 (Biognosys AG) using default settings. PCA analysis based on the protein report table exported from Spectronaut was performed using R version 3.5.0. Data are available via ProteomeXchange with identifier PXD011652.

**Statistical analysis**. Statistical analyses were performed with GraphPad Prism software. Statistical significance was analyzed using two-tailed unpaired Student's *t* test between two different groups. For multiple comparisons, a one-way ANOVA analysis followed by Newman–Keuls post test was performed. Results were considered significant when $p < 0.05$. Data are shown as mean ± SEM unless other specification.

**Reporting summary**. Further information on research design is available in the Nature Research Reporting Summary linked to this article.

## Data availability

The microarray datasets generated during and/or analyzed during the current study are available in the GEO/NCBI (GEO accession: GSE108368). The mass spectrometry proteomics data that support the findings of this study have been deposited in the ProteomeXchange Consortium via the PRIDE[54] partner repository with the dataset identifier PXD011652. The mass spectrometry proteomics (Tandem Mass Tags—TMT) data have been deposited to the ProteomeXchange Consortium via the PRIDE[54] partner repository with the dataset identifier PXD019316. Databases used: Uniprot database (Swissprot entry only, release 2016_01, 16,747 entries); CellAge database (http://genomics.senescence.info/cells/). The authors declare that the data supporting the findings of this study are available within the paper and its supplementary information files. All the figures have associated source data. No restriction is applied to the data presented. Source data are provided with this paper.

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

## Acknowledgements

This work was funded by FEDER through the Program COMPETE and by Portuguese fund through FCT in context of the projects EXPL/BIM-MED/2267/2013 and POCI-01-0145-FEDER-029229, as well as the European project ERAatUC (ref. 669088). PRP wishes to thank FCT for a BD fellowship (SFRH/BD/71042/2010). AR is supported by the Miguel Servet research contract CPII15/00003 from Instituto de Salud Carlos III, Spain. The FLI is a member of the Leibniz Association and is financially supported by the Federal Government of Germany and the State of Thuringia. The authors gratefully acknowledge support from the FLI proteomics core facility. The authors would like to thank Dr Carlos Lopez-Otín for providing the $Lmna^{G609G/+}$ mice.

## Author contributions

P.R.P. and L.F. designed the study, did the literature search and wrote the manuscript. P.R.P. conducted the study. P.R.P., L.E., and H.V. collected the in vitro data. P.R.P., G.C., A.R., A.S.-G., K.H., C.N., and N.L. conducted and analyzed in vivo data. D.T. and J.P.M. processed and analyzed raw genomic data. A.-L.E. and X.N. generated iPSCs from Progeria fibroblasts, provided expertise in the Progeria biology and in the interpretation of the results. L.M., A.B., R.L.S., P.R.P., L.F. and J.C.S. generated the isogenic cell line. T.C. performed pathological evaluation of the tissues. D.S. performed proteomics experiments and analyzed the data with the support of A.O.

## Competing interests

The authors declare no competing interests.
