## [Peer Review File · Nature Communications]

Reviewers' comments:

Reviewer #1 (Remarks to the Author):

This work studied the vulnerability of HGPS iPSC derived SMCs to flow shear stress, and demonstrated that the detachment of HGPS-iPSC SMCs was mediated by the up-regulation of MMP13. In addition, MMP13 inhibitor treatment rescued the SMC loss both in vitro and in vivo.

Concerns :

1. I am concerned about the data qualities. Lots of mistakes exist throughout the manuscript. For example, the Y-axis of Fig 2a.1 is incorrect; the hVSMCs are misspelled as hVMCs in Fig 2b; the legend information of Y-axis in supplementary Fig 11b is missing. High-quality images are needed, including representative pictures of cellular morphology and immunofluorescence.
2. The genetic background of N-iPSC, HGPS-iPSC, HGPS skin fibroblasts and hVSMCs should be addressed. All the phenotypes should be determined using isogenic stem cell models (e.g. after gene correction).
4. In supplementary Fig 1a.1 and b.1, why is progerin expressed in N-iPSC CD34+ cells? What about the protein levels?
5. In supplementary Fig 3b, α -SMA exhibited both cytoplasmic and nuclear location. Why? Very few abnormal nuclei were observed (nuclear deformation is a typical characteristic for HGPS-SMCs or fibroblasts). Please explain.
6. In supplementary Fig 10, DNA damage analysis in N-iPSC SMCs should be included.
7. In supplementary Fig 12, only the effect of flow in a co-culture of HGPS-iPSC SMCs with HUAECs was evaluated. How about co-culture of N-iPSC SMCs with HUAECs? In addition, the proportion of different cell types during the co-culture of SMCs and HUAECs should be provided. How to prove the culture condition used is optimal/suitable to all cell types?
8. Cellular senescence phenotype data should be included, as the author mentioned that 51 differentially expressed genes are associated with cellular senescence.
9. Are progerin and MMP13 included in the differentially expressed protein list? The expression levels of MMP13 in KiHt or KiKo mice should be shown, in order to confirm the MMP13 knockout.
10. While decreased HS was observed in KiKo mice (see Fig 3c), no significant change was seen (see Fig 3g). Please explain.
11. In Fig 4a, the internal elastic lamina and the adventitial border should be determined. How to understand the increased blue staining in KiHt and KiKo? Does it mean that Ki mice had increased thickness of vessel walls?
12. In terms of mechanism, the relationship between progerin, MMP13 and HS needs to be clarified. How about the epistasis of these molecules? Can MMP13 overexpression in N-iPSC-derived SMCs or hVSMCs mimic the phenotypic defects observed in HGPS-iPSC-derived SMCs? Can the vulnerability to flow shear stress be rescued by restoration of functional lamin A/C protein in HGPS-SMCs?

Reviewer #2 (Remarks to the Author):

This manuscript by Pitrez and colleagues, "Vulnerability of Progeroid Smooth Muscle Cells to Biomechanical Force is Mediated by MMP13," combines SMCs derived from HGPS patient iPSCs with a microfluidic device to assess the differential impact of shear stress on these diseases SMCs and investigate possible molecular mechanisms at play. The authors then correlate their findings with mouse models of the disease that also variously knock out alleles of MMP13, their gene of interest in mediating some of the SMC loss seen in HGPS patient aortas.

The work is novel and exciting, with compelling in vitro findings correlated nicely with mouse models. A suitably-revised manuscript would be of great interest and utility to the stem cell, vascular biology, and HGPS research fields. There are some concerns regarding the physiological relevance of some of the studies using their microfluidic device, with some of the studies to be lacking in either appropriate controls or needed additional analyses. Some of the findings are also overstated.

Major Comments:

1. Choice of differentiation protocol: There are multiple published protocols on differentiation of SMCs from iPSCs; these various protocols demonstrate the importance of protocol choice - serum-rich vs. serum-starving media, TGF β , other growth factors - on the synthetic vs. contractile status, the presumed embryologic origin, the ECM production, and the functional behavior of the resulting SMCs. Please discuss the choice of protocol here, including any potential limitations of it with regards to modeling HGPS specifically, and if any other characterization has been performed.
2. Page 4: the authors report >95% of cells expressing calponin, but it is clear from the supplementary data that overall expression of calponin is lower in HGPS cells than in normal cells after both the inductive and maturation media steps. Calponin specifically seems dysregulated in HGPS. Do the authors have any explanation for this difference?
3. Page 6: the authors state "SMC detachment is due to progerin accumulation" but this statement runs counter to discussion elsewhere in the manuscript as well as the fact that HGPS fibroblasts (with high progerin levels) do not experience the same detachment. Progerin accumulation could be causative, but progerin could also be a mediator or confounder in another process (related to focal adhesion assembly, cytoskeletal response to ECM changes, etc.). Can the authors clarify?
4. Page 6: the authors show that SMCs demonstrate "poor proliferation" - but contractile SMCs in general should not be highly proliferative. Is this an expected finding for the authors?
5. The use of capillary-induced seeding of SMCs into the microfluidic device and physiologic shear stresses is novel; however, as the authors admit, exposure of SMCs directly to shear stress is non-physiologic. The authors should perform some of the experiments with an endothelial monolayer in co-culture with the SMCs to strengthen data and show physiologically relevant results. Additionally, please include images of both the confluent EC layer and confluent vSMC layer in the co-culture system before exposing cells to arterial flow conditions. This would better show that EC monolayer does not significantly change despite vSMC detachment.
6. MMP 13 is secreted in an inactive/zymogen form. Could you please discuss the process of activation of this protein and whether analyzing for the active vs. inactive form of the protein would have relevance to their work? This would better implicate this MMP in disease pathogenesis
7. Page 9: the authors report heart rate data for the mice without discussing what is expected of heart rate in HGPS vs. WT. Would one expect heart rate to go down with SMC loss (less vasoactive/contractile, less thick wall media = larger lumen) or to go up (in response to increased resistance from calcification, etc.)? The changes in heart rate seen do not seem clinically significant nor do they seem logically related to SMC loss.
8. In general, there is a wealth of experimental data in the supplementary information that is equally important and compelling to that presented as part of the main manuscript. Given that the authors can show up to 10 figures/tables, it is suggested that the more compelling supplemental data (the HOZ mSMC figure, the osteogenic pathway analysis, the co-culture studies, and conditioned media study) be incorporated into main figures.

9. Page 10: pls add measurements of progerin expression in the Batimastat study. Additionally, in fig 5, "wild type" heart rate is reported, but were wild type mice exposed to BB94? Did their heart rate change?

10. The discussion of heparan sulfate expression and the glycocalyx analyses are incomplete. Would not heparan sulfate also be expected to increase under flow in normal cells if components are, as the authors say "involved in flow shear stress sensing" under normal conditions? Figure 6 largely shows internal comparisons of HGPS under flow at day 4 or 6 vs. an earlier timepoint, but how normal SMCs behave is not shown as part of these analyses. It is important to compare here. Why does heparinase cleavage seem differentially effective in HGPS vs. control SMCs (Supp fig 16)?

11. The authors do not discuss the role of MMP13 in processing other ECM proteins. Does MMP13 process collagen, laminin, fibronectin, etc.? Additionally, consider analyzing integrin subunits expression that could be confounders in the MMP13 and glycocalyx mediated hypothesis. Finally, the authors claim in the discussion that upregulation of MMP13 is "mediated by" the glycocalyx based on their heparinase study, but it seems that this may be an overstatement of their results and the picture could be more complicated.

Minor Comments:

1. Page 6: The authors list Wnt and DNA damage as the pathways of interest in the osteogenic program being turned on; however, the baseline Wnt expression values are much higher in normal vs. HGPS SMCs. Can the authors clarify why they are claiming that these programs are causative here? Do they have any data showing that blocking these pathways abrogates alk phos signaling, for example?

2. Page 6/Supplemental figure 13: The HOZ SMCs are depicted on a different magnification/scale than WT, making them appear hypertrophic. Please correct this in the figure.

3. It would be helpful to see an immunofluorescent stain of the hVSMC-deposited decellularized ECM to see the ECM proteins as well as any residual MMP13, glyocalyx proteins, etc. present.

4. Page 9: the authors refer to a "mutated form" - typo, should be "mutated mouse" (so as not to confuse with a mutated form of MMP13).

5. Figures 3 and 4 - can the authors provide larger/higher resolution image panels of the histology and orcein staining? It is hard to evaluate the authors' claims of SMC loss from the SMA staining, and the elastic fiber layer looks equally thick in the KiWt (though dysregulated/not as compact).

6. The references to Fig 4b and 4c in the text appear incorrect. Additionally, the wording of the discussion of the 25% of proteins expressed "at closer levels" is hard to understand. Please re-word and clarify exactly what comparisons are being made in the text.

7. The authors make reference on page 13 to "patency of SMCs" - is this what is meant? Can the authors clarify or re-word?

Reviewer: 1

1. I am concerned about the data qualities. Lots of mistakes exist throughout the manuscript. For example, the Y-axis of Fig 2a.1 is incorrect; the hVSMCs are misspelled as hVMCs in Fig 2b; the legend information of Y-axis in supplementary Fig 11b is missing. High-quality images are needed, including representative pictures of cellular morphology and immunofluorescence.

We thank the reviewer for bring this to our attention and assure you that we have thoroughly gone through all of the figures to rectify any mistakes and to improve the readability of the figure. Specifically, in the revised version of the manuscript the Y-axis of Fig 4a.1 (previously Fig. 2a.1) was changed from “-log10(p-value)” to “p-value”, the “hVMCs” in Fig. 4b (previously Fig. 2b) was changed to “hVSMCs” and the Y-axis was added to Supplementary Fig.13b (previously Supplementary Fig. 11b). In addition, high-quality images in Supplementary Fig. 4b were added to the revised version of the manuscript.

2. The genetic background of N-iPSC, HGPS-iPSC, HGPS skin fibroblasts and hVSMCs should be addressed. All the phenotypes should be determined using isogenic stem cell models (e.g. after gene correction).

In the revised version of the manuscript, the genetic background of N-iPSC, HGPS-iPSC, HGPS skin fibroblasts and hVSMCs was evaluated and included. The authors have performed LMNA (NM_170707.4 transcript) exon 11 Sanger sequencing in the cell lines and results presented as Supplementary Figure 1. As expected, HGPS-iPSC and HGPS skin fibroblasts presented a HGPS classic mutation (heterozygous c.1814C>T), and N-iPSC and hVSMCs presented a wild type sequence (homozygous c.1824C). In addition, and as requested, the authors have generated a frameshift mutant stem cell line derived from the HGPS-iPSC line to validate the results. This is not an isogenic line *per se* but an attenuated disease line generated and described recently by Carlos Otin and Izpisua Belmonte labs^{1,2}. Importantly, in this line, there is no accumulation of progerin (Fig. 2). Specifically, a guide RNA was used to knockout the HGPS mutant allele. The resulting line (HGPS Δ 2-iPSCs) exhibits a frameshift mutation due to a two-base pair deletion on exon 11, upstream of the HGPS point mutation (1814C>T), as determined by Sanger sequencing (Fig. 2 and Supplementary Fig.10). Notably, HGPS Δ 2-iPSCs did not accumulate progerin upon differentiation as demonstrated at the transcript and protein level (Fig. 2). Upon SMC differentiation, HGPS Δ 2-iPSC-derived SMCs

exhibited an increase in SMC markers, patency under flow conditions and normal MMP13 expression as compared to HGPS-iPSC-derived SMCs.

4. *In supplementary Fig 1a.1 and b.1, why is progerin expressed in N-iPSC CD34+ cells? What about the protein levels?*

The authors would like to clarify the reviewer that the expression of progerin observed in N-iPSC CD34⁺ cells was extremely low (mean CT value = 37,63) and biologically irrelevant. In the revised version of the manuscript, the authors evaluated the expression of progerin in N-iPSC CD34⁺, HGPS-iPSC CD34⁺ and HGPS fibroblasts (positive control) by western blots analyses. Results showed HGPS fibroblasts and HGPS-iPSC-CD34⁺ cells expressed progerin while N-iPSC-SMCs did not. The authors have added the results to Supplementary Fig. 2b.3.

5. *In supplementary Fig 3b, α -SMA exhibited both cytoplasmic and nuclear location. Why? Very few abnormal nuclei were observed (nuclear deformation is a typical characteristic for HGPS-SMCs or fibroblasts). Please explain.*

The accumulation of α -SMA in the cytoplasm and in cell nuclei was a technical artifact of the cell permeabilization process. In the revised version of the manuscript, new representative images for α -SMA, calponin and SMMHC expression were taken by confocal microscopy and added to Supplementary Fig 4b. In addition, the nuclear morphology was also examined in more detail by confocal microscopy. Not all the cells accumulate progerin which decreases the number of cells with dysmorphic nuclei (Supplementary Fig. 4b).

6. *In supplementary Fig 10, DNA damage analysis in N-iPSC SMCs should be included.*

In the revised version of the manuscript, the authors have performed DNA damage analyses in N-iPSCs SMCs (Supplementary Figures 12a.1 and a.2). As expected, the fold change between N-iPSCs SMC cultured under flow conditions and static conditions was similar to the fold change presented by hVSMCs. This means that the flow conditions did not increase the DNA damage in N-iPSCs SMCs or hVSMCs.

7. *In supplementary Fig 12, only the effect of flow in a co-culture of HGPS-iPSC SMCs with HUAECs was evaluated. How about co-culture of N-iPSC SMCs with HUAECs? In addition, the proportion of different cell types during the co-culture of SMCs*

and HUAECs should be provided. How to prove the culture condition used is optimal/suitable to all cell types?

In the revised version of the manuscript, the authors have performed additional experiments to: (i) evaluate the effect of flow in a co-culture of N-iPSC SMCs with HUAECs (Supplementary Fig. 15), (ii) evaluate the effect of the EC:SMC ratio's in the final vulnerability of SMCs (Supplementary Fig.15) and (iii) prove that the culture conditions used were suitable to all cell types (Supplementary Figure 14). The following information was added to the manuscript (pages 6-7): "Initially, we screened different culture conditions and we found that endothelial growth media-2 (EGM2) medium was a suitable medium to support both cells (Supplementary Fig.14). Then, we co-cultured HUAECs and HGPS-iPSC SMCs at different ratios (1.6, 1 and 0.6) under flow conditions. In all the ratios tested, we had a monolayer of HUAECs (Supplementary Fig. 15a) and HGPS-iPSC SMCs (data not shown) or N-iPSC SMCs at time zero. After 6 days in flow conditions, a significant percentage (>40%) of HGPS-iPSC SMCs was lost (Supplementary Fig. 15b). For the highest ratio tested (1.6), the loss of HGPS-iPSC SMCs occurred without visible loss of ECs. Yet, for EC:SMC ratios below 1, part of ECs also detached from the microfluidic chamber indicating that a low EC density may turn ECs vulnerable to flow conditions. Importantly, cell vulnerability to flow conditions was only observed in co-cultures of HGPS-iPSC SMCs but not N-iPSC SMCs (Supplementary Fig. 15c)."

8. Cellular senescence phenotype data should be included, as the author mentioned that 51 differentially expressed genes are associated with cellular senescence.

The authors have characterized HGPS-iPSC SMCs and N-iPSC SMCs for p21 and SA- β -galactosidase. Our results showed that HGPS-iPSC SMCs had higher levels of both indicators than N-iPSC SMCs (Supplementary Fig. 16a). In addition, the authors have performed proteomic analyses of HGPS-iPSC SMCs at days 0 and 4, using data independent mass spectrometry ^{3,4}, to investigate whether the level of cell senescence increases overtime (Supplementary Fig. 16b). Our results showed that 1170 proteins were differentially expressed at day 4 relatively to day 0 ($\log_2FC \geq 1$; $p < 0.05$). From these proteins, 33 were associated with cellular senescence, as determined by the intersection of all the differentially expressed proteins (1170) with the CellAge database (279 genes) (Table S8). Therefore, the level of SMC senescence increases after culture in flow conditions. The following information was added to the revised version of the manuscript (page 8): "At the protein

levels, HGPS-iPSC SMCs expressed higher levels of p21 and SA- β -galactosidase than N-iPSCs-SMCs (Supplementary Fig. 16a) and the level of senescence markers increased after culture of HGPS-iPSC SMCs in flow conditions (Supplementary Fig. 16b).”

9. Are progerin and MMP13 included in the differentially expressed protein list? The expression levels of MMP13 in KiHt or KiKo mice should be shown, in order to confirm the MMP13 knockout.

The reviewer makes a valid point and we have amended the text to clarify why neither progerin nor MMP13 are included in the differentially expressed protein list. Progerin is a mutated protein and thus not identified by the mass spectrometry. MMP13 is a secreted protein and the levels in cells were not detectable by mass spectrometry. Both information's was added to the caption of Fig. 6. Nevertheless, levels of progerin and MMP13 were analyzed by immunofluorescence (Fig. 5c) and ELISA (Fig. 5b), respectively, as well as by qRT-PCR. The expression levels of MMP13 in KiHt and KiKo mice were added in the revised version of the manuscript to confirm the MMP13 knockout (Supplementary Fig. 20).

10. While decreased HS was observed in KiKo mice (see Fig 3c), no significant change was seen (see Fig 3g). Please explain.

Indeed, a decrease in heparan sulfate intensity is observed in *Lmna*^{G609G/G609G}*Mmp13*^{-/-} mice relatively to *Lmna*^{G609G/G609G}*Mmp13*^{+/+} mice (now Fig. 5g); however, no statistical significance was observed. The decrease was not significant likely because the up-regulation of heparan sulfate was not observed in *Lmna*^{G609G/G609G}*Mmp13*^{+/+} mice as seen in other progeroid animal models ^{5,6} or in HGPS patients ⁷. Therefore, further time (above 10 weeks) is likely needed to observe a statistical difference in the expression of heparan sulfate between *Lmna*^{G609G/G609G}*Mmp13*^{+/+} mice and *Lmna*^{G609G/G609G}*Mmp13*^{-/-} mice. The following note was added to the manuscript (page 16): “Since the upregulation of heparan sulfate was not observed in *Lmna*^{G609G/G609G}*Mmp13*^{+/+} mice, it is not surprising that we could not observe a statistical decrease in heparan sulfate in *Lmna*^{G609G/G609G}*Mmp13*^{-/-} mice.”

11. In Fig 4a, the internal elastic lamina and the adventitial border should be determined. How to understand the increased blue staining in KiHt and KiKo? Does it mean that Ki mice had increased thickness of vessel walls?

In the revised version of the manuscript, the authors have defined the internal elastic lamina and the adventitial border in Figure 6a (previous Fig. 4a). Arrows were included in the figure (black arrow = internal elastic lamina; white arrow = adventitial border), to delimitate the border of the internal elastic lamina and the adventitial. Materials and methods explaining the procedure were added to the supplementary material. The blue staining was given by the methyl blue, used as counterstain in the Orcein elastic staining method. Its intensity may vary due to external factors, like the time to fixation (post-mortem) and also internal factors, like the density and affinity of the extracellular matrix components to these chromogens; hence, these variations should not be used as proxy for vessel wall thickness.

12. In terms of mechanism, the relationship between progerin, MMP13 and HS needs to be clarified. How about the epistasis of these molecules? Can MMP13 overexpression in N-iPSC-derived SMCs or hVSMCs mimic the phenotypic defects observed in HGPS-iPSC-derived SMCs? Can the vulnerability to flow shear stress be rescued by restoration of functional lamin A/C protein in HGPS-SMCs?

In the previous version of the manuscript, the authors had shown that conditioned media collected from HGPS-iPSC SMCs in flow conditions for 4 days could induce the detachment of flow shear stress-insensitive hVSMCs (Fig. 4c). Now, in the revised version of the manuscript, the authors have overexpressed MMP13 in hVSMCs and cultured the modified cells in flow culture conditions for 7 days (Supplementary Fig. 19). At the end, MMP13-overexpressing cells were in low number as compared to wild type cells. Therefore, these results suggest that the overexpression of MMP13 in hVSMCs mimicked the phenotypic defects observed in HGPS-iPSC-derived SMCs. The authors have also knockout specifically the HGPS mutant allele. The resulting line (HGPS Δ 2-iPSCs; see point 2) did not accumulate progerin upon SMC differentiation as demonstrated at the transcript and protein level (Fig. 2), resisted to flow conditions and expressed lower levels of MMP13 as compared to HGPS-iPSC-derived SMCs. Overall, our results indicate that HGPS-iPSC SMC detachment can be rescued by the correction of the HGPS mutant allele.

Reviewer: 2

1. Choice of differentiation protocol: There are multiple published protocols on differentiation of SMCs from iPSCs; these various protocols demonstrate the importance of protocol choice - serum-rich vs. serum-starving media, TGF β , other growth factors - on the synthetic vs. contractile status, the presumed embryologic origin, the ECM production, and the functional behavior of the resulting SMCs. Please discuss the choice of protocol here, including any potential limitations of it with regards to modeling HGPS specifically, and if any other characterization has been performed.

The authors have followed the suggestion of the reviewer and added the following information to the manuscript: “Multiple protocols have been described in the literature for the differentiation of iPSCs into SMCs, either via an intermediate progenitor stage or directed differentiation^{14,42-44}. These protocols are highly variable in terms of SMC differentiation efficiency, timescale and functionality (non-dividing contractile phenotype *versus* proliferative phenotype, secretory profile), likely due to the choice of precursor population to derive the SMC subtypes, the chemical composition of the differentiation medium, as well as the choice of inductive SMC factors (e.g. PDGF-BB, TGF- β 1, retinoic acid). Three previous studies have reported the differentiation of HGPS iPSCs into SMCs^{7,9,45} by direct differentiation⁷ or by using an intermediate progenitor (i.e. mesenchymal stem cells⁴⁵ or CD34⁺ cells⁹). In some cases, SMCs were not terminally differentiated (as confirmed by the expression of SMMHC)⁷, in others the percentage of SMCs was relatively low (i.e. only 50-60% of the differentiated cells showed specific SMC markers including α -SMA, calponin 1 and SMMHC)⁴⁵ and no indication of SMC functionality⁹ (e.g. contractility, intracellular accumulation of calcium after exposure to vasoactive agents) was reported. In the present study, we showed that the differentiation of HGPS-iPSCs induces the activation of the NOTCH signaling pathway, a hallmark of progerin-expressing cells¹⁵. This is observed in the CD34⁺ progenitor cells and after their differentiation into SMCs. The CD34⁺ cells have been reported to express KDR and CD31⁴³ and, thus, are likely of lateral plate mesoderm origin^{42,44}. Importantly, the differentiated cells express high levels of all the SMC markers analyzed (α -SMA, calponin and SMMHC), are contractile in response to the muscarinic receptor agonist carbachol as observed in typical human aortic SMCs, and when matured in culture for approximately 30 days they express progerin. Therefore, our differentiation protocol compares favorably to other protocols in term of SMC yield and functionality. Interestingly,

HGPS-iPSC SMCs express lower levels of calponin than in N-iPSC SMCs but the reason and possible implications behind this phenotypic difference remain to be determined. Nevertheless, most of the HGPS-iPSC SMCs expressed calponin at the protein level, both at the induction and maturation steps (Supplementary Figs. 3 and 4). A previous study has reported heterogeneous sized calponin 1-staining inclusion bodies in the cytoplasm of HGPS-SMCs⁹; however, such structures were not observed in the current study.”

2. Page 4: the authors report >95% of cells expressing calponin, but it is clear from the supplementary data that overall expression of calponin is lower in HGPS cells than in normal cells after both the inductive and maturation media steps. Calponin specifically seems dysregulated in HGPS. Do the authors have any explanation for this difference?

In the revised version of the manuscript, the authors have added the following information to the discussion section (pages 14-15). “Interestingly, HGPS-iPSC SMCs express lower levels of calponin than in N-iPSC SMCs but the reason and possible implications behind this phenotypic difference remain to be determined. Nevertheless, most of the HGPS-iPSC SMCs expressed calponin at the protein level, both at the induction and maturation steps (Supplementary Figs. 3 and 4). A previous study has reported heterogeneous sized calponin 1-staining inclusion bodies in the cytoplasm of HGPS-SMCs⁹; however, such structures were not observed in the current study.”

3. Page 6: the authors state “SMC detachment is due to progerin accumulation” but this statement runs counter to discussion elsewhere in the manuscript as well as the fact that HGPS fibroblasts (with high progerin levels) do not experience the same detachment. Progerin accumulation could be causative, but progerin could also be a mediator or confounder in another process (related to focal adhesion assembly, cytoskeletal response to ECM changes, etc.). Can the authors clarify?

In the revised version of the manuscript, the authors have clarified the statement to “Our results indicate that SMC detachment is mediated by progerin accumulation (...).” Indeed, it is unlikely that progerin accumulation is the causative of cell detachment because HGPS fibroblasts accumulate high levels of progerin and did not detach from the microchannels. It is likely that the accumulation of progerin mediates the detachment process and further investigation is needed to unravel the relationship between progerin and MMP13. So far, our results indicate that inhibition of progerin in HGPS-iPSC SMC by antisense morpholinos significantly decreased HGPS-iPSC SMC detachment. In addition, the knockout of the HGPS

mutant allele in HGPS-SMCs prevented cell detachment after culture in flow conditions. The following note was added to the discussion section (page 15): “In addition, we found that the accumulation of progerin is a mediator and not the cause of SMC detachment because HGPS fibroblasts accumulate high levels of progerin and do not detach in flow conditions. Yet, both inhibition of progerin by morpholinos and the knockout of the HGPS mutant allele in HGPS-SMCs decreased or prevented SMC detachment in flow culture conditions.”

4. *Page 6: the authors show that SMCs demonstrate “poor proliferation” - but contractile SMCs in general should not be highly proliferative. Is this an expected finding for the authors?*

In the revised version of the manuscript, the authors have clarified the meaning of poor proliferation: “HGPS-iPSC SMC detachment does not seem to be mediated by cell apoptosis. Before cell detachment, HGPS-iPSC SMCs showed: (i) poor proliferation (as monitored by Ki67 staining) confirming their contractile phenotype (Fig. 1h), (...)”. Indeed, the poor proliferation of N-iPSC SMCs and HGPS-iPSC SMCs is an expected finding considering their contractile phenotype.

5. *The use of capillary-induced seeding of SMCs into the microfluidic device and physiologic shear stresses is novel; however, as the authors admit, exposure of SMCs directly to shear stress is non-physiologic. The authors should perform some of the experiments with an endothelial monolayer in co-culture with the SMCs to strengthen data and show physiologically relevant results. Additionally, please include images of both the confluent EC layer and confluent vSMC layer in the co-culture system before exposing cells to arterial flow conditions. This would better show that EC monolayer does not significantly change despite vSMC detachment.*

In the revised version of the manuscript, the authors have performed additional experiments to: (i) evaluate the effect of flow in a co-culture of N-iPSC SMCs with HUAECs (Supplementary Fig. 15), (ii) evaluate the effect of the EC:SMC ratio's in the final vulnerability of SMCs (Supplementary Fig.15) and (iii) prove that the culture conditions used were suitable to all cell types (Supplementary Figure 14). The following information was added to the manuscript (pages 6-7): “Initially, we have screened different culture conditions and we found that EGM2 medium was a suitable medium to support both cells (Supplementary Fig.14). Then, we co-cultured HUAECs and HGPS-iPSC SMCs at different

ratio's (1.6, 1 and 0.6) under flow conditions. In all the ratio's tested, we had a monolayer of HUAECs (Supplementary Fig. 15a) and HGPS-iPSC SMCs (data not shown) or N-iPSC SMCs at time zero. After 6 days in flow conditions, a significant percentage (>40%) of HGPS-iPSC SMCs was lost (Supplementary Fig. 15b). For the highest ratio tested (1.6), the loss of SMCs occurred without visible loss of ECs. Yet, for EC:SMC ratio's below 1, part of ECs also detached from the microfluidic chamber indicating that a low EC density may turn ECs vulnerable to flow conditions. Importantly, cell vulnerability to flow conditions was only observed in co-cultures of HGPS-iPSC SMCs but not N-iPSC SMCs (Supplementary Fig. 15c)."

6. *MMP 13 is secreted in an inactive/zymogen form. Could you please discuss the process of activation of this protein and whether analyzing for the active vs. inactive form of the protein would have relevance to their work? This would better implicate this MMP in disease pathogenesis.*

The authors would like to thank the reviewer for the comment. The authors have performed additional experiments to elucidate whether the MMP13 secreted by HGPS-iPSC SMCs under flow conditions was active or inactive. The results are now presented as Supplementary Figure 19. The following information was added to the manuscript (pages 8-9): "Because MMP13 is produced by cells in an inactive form (proMMP13) which is then activated by cell membrane MMPs, namely MMP14 (also called MT1-MMP) and MMP2 (also called gelatinase A) ¹⁵, the catalytic activity of MMP13 secreted by HGPS-iPSC SMCs was analyzed (Supplementary Fig. 19). The concentration of proMMP13 and active MMP13 increased approximately 8- and 5-fold, respectively, in culture media of HGPS-iPSC SMCs cultured in flow conditions from day 0 to day 4. Moreover, the concentration of proMMP13 and active MMP13 in cell culture media collected from N-iPSC SMCs cultured in flow conditions for 4 days was more than 4-fold lower than the one observed with HGPS-iPSC SMCs."

7. *Page 9: the authors report heart rate data for the mice without discussing what is expected of heart rate in HGPS vs. WT. Would one expect heart rate to go down with SMC loss (less vasoactive/contractile, less thick wall media = larger lumen) or to go up (in response to increased resistance from calcification, etc.)? The changes in heart rate seen do not seem clinically significant nor do they seem logically related to SMC loss.*

Heart rate is a parameter influenced by many factors (vascular features and intrinsic cardiac features, including for instance alterations of the cardiac cells' gap junction protein Cx43)¹⁶ in opposite fashions, as the reviewer correctly pinpointed, and indeed, it's not necessarily correlated with vascular SMC counts. Nevertheless, it was chosen as one of the parameters that can help determine the overall health status of the HGPS model and thus the efficacy of a treatment, given that bradycardia was a clinical anomaly evidenced in this model and also Zmpste 24^{-/-} progeria mouse model^{16,17}. In the BB94 experiment, heart rates may have not been fully corrected because of the incomplete MMP13 inactivation obtained with the oral administration of the drug compared to KO of the *Mmp13* gene, while vSMC counts were corrected in both conditions and even upon partial MMP13 inactivation by BB94. The following information was added to the manuscript (page 11): "Heart rate was chosen as a measure of the overall health status of the HGPS model and the derived double mutant lines, given that bradycardia was a clinical abnormality evidenced in both LmnaG609G/G609G mouse as well as Zmpste 24^{-/-} progeria mouse models^{16,17}."

8. *In general, there is a wealth of experimental data in the supplementary information that is equally important and compelling to that presented as part of the main manuscript. Given that the authors can show up to 10 figures/tables, it is suggested that the more compelling supplemental data (the HOZ mSMC figure, the osteogenic pathway analysis, the co-culture studies, and conditioned media study) be incorporated into main figures.*

The authors have included an additional image in the main manuscript related to the generation of isogenic cell line (now Fig. 2) and the results related to HOZ mSMCs (now Fig. 3). The co-culture studies were presented as supplementary information (Supplementary Figs. 14 and 15) since the authors collected a significant amount of new data. The osteogenic differentiation analyses in HGPS-iPSC SMCs were performed in order to demonstrate that the cells recapitulated aspects reported previously for HGPS-mSMCs and thus were not included as a main figure.

9. *Page 10: pls add measurments of progerin expression in the Batimastat study. Additionally, in fig 5, "wild type" heart rate is reported, but were wild type mice exposed to BB94? Did their heart rate change?*

In the revised version of the manuscript, the authors have added progerin expression in the Batimastat study (Supplementary Fig. 20c). No differences were observed in progeria mice

treated with Batimastat compared to progeria mice that received placebo. In Figure 7 (previously Figure 5), wild type mice were not exposed to BB94. The authors have added the following clarification to the legend of Figure 7: “Wild type mice were not exposed to BB94”. The authors have not measured the heart rate of wild type mice exposed to BB94. Yet, the authors would not expect a change in mice heart rate since the drug has passed successfully phase I clinical trials and thus showed cardio safety.

10. The discussion of heparan sulfate expression and the glycocalyx analyses are incomplete. Would not heparan sulfate also be expected to increase under flow in normal cells if components are, as the authors say “involved in flow shear stress sensing” under normal conditions? Figure 6 largely shows internal comparisons of HGPS under flow at day 4 or 6 vs. an earlier timepoint, but how normal SMCs behave is not shown as part of these analyses. It is important to compare here. Why does heparinase cleavage seem differentially effective in HGPS vs. control SMCs (Supp fig 16)?

In the revised version of the manuscript, the authors have shown that syndecan 2 gene, which encodes the transmembrane (type I) heparan sulfate proteoglycan, is indeed upregulated at day 4 relatively to day 0 in hVSMCs or N-iPSC SMCs (Supplementary Fig. 22); however, this up-regulation did not translate in a high expression of the protein. In addition, the number of mRNA MMP13 transcripts also increased at day 4 relatively to day 0, and thus our results indicate that normal SMCs respond to flow. Taking into account gene expression results, the glycocalyx produced by HGPS-iPSC SMCs seems different from the one produced by N-iPSC SMCs or hVSMCs. This might translate into variable catalytic actions by heparinase. The following information was added to the manuscript (page 12): “The glycocalyx is a surface layer of proteoglycans and glycosaminoglycans that are immobilized in the cell membrane. Glycocalyx components have been shown to be involved in flow shear stress sensing by SMCs^{18,19}. To identify the mechanism underlying the up-regulation of MMP13 in HGPS-iPSC-SMCs cultured under arterial flow, we analyzed glycocalyx gene mRNA transcripts (Fig. 8b). Interestingly, glycocalyx transcripts were up-regulated in HGPS-iPSC SMCs cultured under flow conditions for 4 days (Fig. 8b). From these upregulated genes, syndecan 2 gene (SDC2), which encodes the transmembrane (type I) heparan sulfate proteoglycan, was also upregulated in hVSMCs or N-iPSC SMCs cultured for 4 days in flow conditions (Supplementary Fig. 21). Because not all the glycocalyx mRNA transcripts were upregulated in hVSMCs and N-iPSC SMCs, the results suggest that the composition of

glycocalyx is likely different in these cells when compared to HGPS-iPSC SMCs. Next, we analyzed the expression of heparan sulfate at the protein level.”

11. *The authors do not discuss the role of MMP13 in processing other ECM proteins. Does MMP13 process collagen, laminin, fibronectin, etc.? Additionally, consider analyzing integrin subunits expression that could be confounders in the MMP13 and glycocalyx mediated hypothesis. Finally, the authors claim in the discussion that upregulation of MMP13 is “mediated by” the glycocalyx based on their heparinase study, but it seems that this may be an overstatement of their results and the picture could be more complicated.*

The authors have performed proteomic analyses of HGPS-iPSC SMCs at days 0 and 4, using data independent mass spectrometry^{3,4} to identify potential ECM targets of the MMP13. Our results indicate that collagen 1A1, preferentially expressed by SMCs (as compared to ECs), might be a potential target of MMP13. The following information was added to the manuscript (page 12-13): “To further investigate a potential ECM target of MMP13 in SMCs, we monitored the expression of ECM components in hVSMCs, HUAECs, N-iPSC SMCs and HGPS-iPSC SMCs. Our results indicate that hVSMCs express higher levels of mRNA that encode collagen 1A1, collagen 3A1, collagen 4A2 and collagen 6A3 than HUAECs (Supplementary Fig. 21c). It has been shown that MMP13 degrades very efficiently the native helix of all fibrillary collagens, including collagen type I²⁰. Our proteomic results indicate that indeed collagen 1A1 is upregulated in HGPS-iPSC SMCs exposed to flow conditions (Supplementary Fig. 16) and thus it may be a potential target for MMP13.”

Minor Comments:

1. *Page 6: The authors list Wnt and DNA damage as the pathways of interest in the osteogenic program being turned on; however, the baseline Wnt expression values are much higher in normal vs. HGPS SMCs. Can the authors clarify why they are claiming that these programs are causative here? Do they have any data showing that blocking these pathways abrogates alk phos signaling, for example?*

The authors have removed the claim and the results presented in Supplementary Fig. 11, in the previous version of the manuscript. The authors maintained the DNA damage results but altered the text of the main manuscript.

2. Page 6/Supplemental figure 13: The HOZ SMCs are depicted on a different magnification/scale than WT, making them appear hypertrophic. Please correct this in the figure.

In the revised version of the manuscript, the authors have followed the suggestion of the reviewer and have replaced the previous figures by new ones with the same magnification/scale.

3. It would be helpful to see an immunofluorescent stain of the hVSMC-deposited decellularized ECM to see the ECM proteins as well as any residual MMP13, glyocalyx proteins, etc. present.

The authors have performed additional experiments to characterize the hVSMC-deposited decellularized ECM. In the revised manuscript (Supplementary Figure 18), the authors demonstrate that the decellularized ECM expressed fibronectin, collagen I and heparin sulfate.

4. Page 9: the authors refer to a “mutated form” - typo, should be “mutated mouse” (so as not to confuse with a mutated form of MMP13).

The authors have followed the recommendation of the reviewer and replaced “mutated form” by “mutated mice”.

5. Figures 3 and 4 - can the authors provide larger/higher resolution image panels of the histology and orcein staining? It is hard to evaluate the authors’ claims of SMC loss from the SMA staining, and the elastic fiber layer looks equally thick in the KiWt (though dysregulated/not as compact).

The authors have included in the revised version of the manuscript larger/higher resolution of the orcein-stained aorta of wild type and mutant mice (now Figure 6; previously Figure 4). The images represent the morphological changes of aortic media tunica in these animals rather than a reduction in the thickness. This issue was clarified in the caption of Figure 6. Regarding Figure 5 (previously Figure 3), it was difficult to include larger magnification images of the panel due to space limitations.

6. The references to Fig 4b and 4c in the text appear incorrect. Additionally, the wording of the discussion of the 25% of proteins expressed “at closer levels” is hard to

understand. Please re-word and clarify exactly what comparisons are being made in the text.

The authors have re-word the sentence to “(...) approximately 25% of the proteins had similar expression in LmnaG609G/G609GMmp13+/- mice and wild type mice (Fig. 6c).”

7. The authors make reference on page 13 to “patency of SMCs” - is this what is meant? Can the authors clarify or re-word?

The authors have re-word the sentence to: “Overall, our study demonstrates that the control of MMP13 expression decreases the vulnerability of SMCs in large vessels and this strategy may be of potential value to reduce the impact of the disease in Progeria patients.”

References:

1. Santiago-Fernandez, O., *et al.* Development of a CRISPR/Cas9-based therapy for Hutchinson-Gilford progeria syndrome. *Nat Med* **25**, 423-426 (2019).
2. Beyret, E., *et al.* Single-dose CRISPR-Cas9 therapy extends lifespan of mice with Hutchinson-Gilford progeria syndrome. *Nat Med* **25**, 419-422 (2019).
3. Buczak, K., *et al.* Spatial Tissue Proteomics Quantifies Inter- and Intratumor Heterogeneity in Hepatocellular Carcinoma (HCC). *Mol Cell Proteomics* **17**, 810-825 (2018).
4. Heinze, I., *et al.* Species comparison of liver proteomes reveals links to naked mole-rat longevity and human aging. *BMC Biol* **16**, 82 (2018).
5. Varga, R., *et al.* Progressive vascular smooth muscle cell defects in a mouse model of Hutchinson-Gilford progeria syndrome. *Proceedings of the National Academy of Sciences of the United States of America* **103**, 3250-3255 (2006).
6. Capell, B.C., *et al.* Inhibiting farnesylation of progerin prevents the characteristic nuclear blebbing of Hutchinson-Gilford progeria syndrome. *Proceedings of the National Academy of Sciences of the United States of America* **102**, 12879-12884 (2005).
7. Olive, M., *et al.* Cardiovascular pathology in Hutchinson-Gilford progeria: correlation with the vascular pathology of aging. *Arterioscler Thromb Vasc Biol* **30**, 2301-2309 (2010).
8. Cheung, C., Bernardo, A.S., Pedersen, R.A. & Sinha, S. Directed differentiation of embryonic origin-specific vascular smooth muscle subtypes from human pluripotent stem cells. *Nat Protoc* **9**, 929-938 (2014).
9. Ferreira, L.S., *et al.* Vascular progenitor cells isolated from human embryonic stem cells give rise to endothelial and smooth muscle like cells and form vascular networks in vivo. *Circ Res* **101**, 286-294 (2007).
10. Vazao, H., das Neves, R.P., Graos, M. & Ferreira, L. Towards the maturation and characterization of smooth muscle cells derived from human embryonic stem cells. *PLoS One* **6**, e17771 (2011).
11. Maguire, E.M., Xiao, Q. & Xu, Q. Differentiation and Application of Induced Pluripotent Stem Cell-Derived Vascular Smooth Muscle Cells. *Arterioscler Thromb Vasc Biol* **37**, 2026-2037 (2017).
12. Liu, G.H., *et al.* Recapitulation of premature ageing with iPSCs from Hutchinson-Gilford progeria syndrome. *Nature* **472**, 221-225 (2011).
13. Zhang, J., *et al.* A human iPSC model of Hutchinson Gilford Progeria reveals vascular smooth muscle and mesenchymal stem cell defects. *Cell Stem Cell* **8**, 31-45 (2011).
14. Zhang, H., Xiong, Z.M. & Cao, K. Mechanisms controlling the smooth muscle cell death in progeria via down-regulation of poly(ADP-ribose) polymerase 1. *Proceedings of the National Academy of Sciences of the United States of America* **111**, E2261-2270 (2014).

15. Knauper, V., *et al.* Cellular mechanisms for human procollagenase-3 (MMP-13) activation. Evidence that MT1-MMP (MMP-14) and gelatinase a (MMP-2) are able to generate active enzyme. *J Biol Chem* **271**, 17124-17131 (1996).
16. Rivera-Torres, J., *et al.* Cardiac electrical defects in progeroid mice and Hutchinson-Gilford progeria syndrome patients with nuclear lamina alterations. *Proceedings of the National Academy of Sciences of the United States of America* **113**, E7250-E7259 (2016).
17. Osorio, F.G., *et al.* Splicing-directed therapy in a new mouse model of human accelerated aging. *Sci Transl Med* **3**, 106ra107 (2011).
18. Shi, Z.D., Wang, H. & Tarbell, J.M. Heparan sulfate proteoglycans mediate interstitial flow mechanotransduction regulating MMP-13 expression and cell motility via FAK-ERK in 3D collagen. *PLoS One* **6**, e15956 (2011).
19. Ainslie, K.M., Garanich, J.S., Dull, R.O. & Tarbell, J.M. Vascular smooth muscle cell glycocalyx influences shear stress-mediated contractile response. *J Appl Physiol (1985)* **98**, 242-249 (2005).
20. Knauper, V., *et al.* The role of the C-terminal domain of human collagenase-3 (MMP-13) in the activation of procollagenase-3, substrate specificity, and tissue inhibitor of metalloproteinase interaction. *J Biol Chem* **272**, 7608-7616 (1997).

REVIEWERS' COMMENTS

Reviewer #1 (Remarks to the Author):

My concerns have been addressed.

Reviewer #2 (Remarks to the Author):

The authors addressed all my comments.